# Direct Neural Network Training on Securely Encoded Datasets

## Abstract

In fields where data privacy and secrecy are critical, such as healthcare and business intelligence, security concerns have reduced the availability of data for neural network training. A recently developed technique securely encodes training, test, and inference examples with an aggregate non-orthogonal and nonlinear transformation that consists of steps of random padding, random perturbation, and random orthogonal matrix transformation, enabling artificial neural network (ANN) training and inference directly on encoded datasets. Here, the performance characteristics and privacy aspects of the method are presented. The individual transformations of the method, when applied alone, do not significantly reduce validation accuracy with fully-connected ANNs. Training on datasets transformed by sequential padding, perturbation, and orthogonal transformation results in slightly lower validation accuracies than those seen with unmodified control datasets, with no difference in training time seen between transformed and control datasets. The presented methods have implications for machine learning in fields requiring data security.

**Keywords**: secure machine learning, data security, encoded training data, encoded test data, non-orthogonal transformation, nonlinear transformation, artificial neural networks.

## 1 Introduction

Privacy and data security are paramount in areas such as healthcare and business intelligence. In healthcare, as an illustrative example, complex regulatory structures exist in different jurisdictions that were designed to protect the privacy of data that might identify an individual and their personal information (Theodos & Sittig, 2020; Phillips, 2018). The data security requirements in these areas have created demand for new technologies that can securely encode data for machine learning training and inference using cloud resources.

Massive amounts of data remain siloed in hospital and other systems because of privacy, data security, and regulatory concerns (Kostkova et al., 2016), and these data remain inaccessible to the cloud resources and technical expertise needed for modern neural network training. The existing barriers to cloud-based machine learning and collaboration using this wealth of healthcare data are particularly important in light of recent dramatic results that have been obtained by training neural networks on the limited number of datasets that have been available to date. For example, deep network training on retinal photographs from diabetic screening programs yielded an accuracy for the detection of diabetic retinopathy that is on par with board-certified ophthalmologists (Gulshan et al., 2016). In addition, use of the same retinal imaging datasets facilitated training of neural networks to make a range of surprisingly accurate inferences related to blood pressure, smoking history, age, and gender (Poplin et al., 2018). The same group more recently trained neural networks on external eye photographs, and these networks could accurately predict levels of the laboratory measures hemoglobin A1C (a measure of diabetes mellitus and its control) and serum cholesterol (Babenko et al., 2022), neither of which were previously known to be associated with any features visible in external photographs of the eye.

There should be an important role for neural networks to dramatically enhance clinical diagnosis (Poplin et al., 2018; Babenko et al., 2022), but machine learning techniques have yet to be widely adopted across healthcare. While the slow pace of adoption may be in part related to a slow-to-adapt culture in healthcare and a complex regulatory environment, data siloing has certainly limited the availability of data and created barriers to transmitting private data to cloud resources for training and inference. Solutions are therefore needed to securely encode data in healthcare and other fields such that encoded data may be directly used in machine learning applications.

An ideal encoding method to establish security for machine learning systems would securely encode training, test, and inference data in a fashion that allows existing architectures to be used on the encoded data, with minimal loss of accuracy and comparable training times. Such an implementation would also require test and inference examples to be similarly encoded to training examples, which would have the benefit of maintaining the business value of training data in collaborations.

Here, the performance of a novel method of secure data encoding for direct use with neural networks (Anonymized, 2022) is described and tested. The method involves a combination of different forms of padding, perturbation, and orthogonal transformation (e.g., fixed shuffling) to create an aggregate non-orthogonal and nonlinear transformation that is applied to training, test, and inference data, with neural networks trained directly on the encoded data. We find that training a fully-connected ANN on data encoded by this aggregate transformation achieves a level of validation accuracy that is similar to results obtained when training with an unencoded dataset.

## 2 Threat model

The encoding transformations described in detail below are designed to protect data privacy in the context of a specific threat model, as shown in Figure 1.

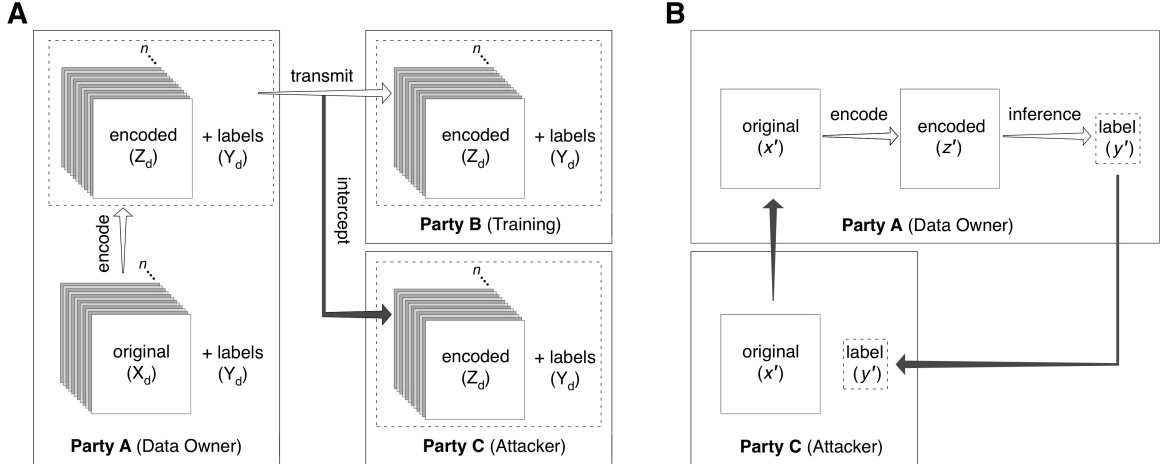

Figure 1: Threat model.
A. *Training stage.* Party A (the data owner) possesses the original dataset ($Xd$) and set of labels ($Yd$). Party A generates an encoded dataset ($Zd$), which is transmitted along with $Yd$ to Party B (ANN training). Party C (attacker) intercepts transmission of *Zd* and *Yd.*
B. *Inference phase.* Party C (attacker) submits an original (unencoded) example ($x'$) that is *not* a member of the original dataset (Xd) from training. Party A (data owner) receives ($x'$) and *privately* encodes it to ($z'$) with the same algorithm used during training. Inference performed on ($z'$) using the trained neural network yields label ($y'$), which is returned to Party C. Only the inference result label ($y'$) is transmitted back to Party C, and after inference is performed, Party A destroys the encoded example ($z'$).

As illustrated in Figure 1A, during training, a data owner (Party A) possesses an original dataset (*Xd*) with *n* examples and a corresponding set of *n* labels (*Yd*). Party A applies the random functions described below to generate an encoded dataset (*Zd*) with the same *n* number of examples and an unchanged set of *n* labels (*Yd*). Party A transmits *Zd* and *Yd* to Party B for performance of ANN training, and Party B uses *Zd* and *Yd* to train a neural network (Figure 1A).

At this stage, we assume that Party C is able to intercept *Xd* and *Yd* during transmission (although, in practice, this configuration would allow Party A and Party B to encrypt *Zd* and *Yd* for transmission by standard 2-key encryption methods). An alternative scenario within the threat model is that Party C might be affiliated with Party B and thus able to obtain *Zd* and *Yd* without a need to intercept the data during transmission. In either scenario, Party C will have access to the neural network trained using *Zd* and *Yd*, either because they are affiliated with Party B and have access to the network trained by Party B, or because they intercepted *Zd* and *Yd* and proceeded to train their own neural network using the encoded data (*Zd*) and associated labels (*Yd*).

As illustrated in Figure 1B, during inference, an attacker (Party C) may present, as a user of software designed to perform inference, an original data example ($x'$) to Party A. Party A then *privately* encodes this example ($x'$) using the same encoding process used for training to yield an encoded example ($z'$). Because Party C does not have access to original examples in the training dataset (*Xd*), this original example ($x'$) presented by Party C is *not* a member of *Xd* and must be generated for this purpose by Party C *without knowledge of the original examples* in *Xd*. In Party A's possession, the encoded example ($z'$) is presented to the trained neural network (obtained from Party B after the training process) to yield a label category prediction ($y'$). Party A returns the label category prediction ($y'$) to the user (in this scenario, the attacker, Party C), *without disclosing the encoded example* ($z'$), and Party A destroys the encoded example ($z'$) after inference is performed.

From these two scenarios, as illustrated in Figure 1, we obtain the threat model, in which an an attacker (Party C) can come to possess the following:

- An encoded training dataset (*Zd*), with *n* encoded examples, and *n* corresponding labels (*Yd*).
- A trained neural network, obtained by training on *Zd* and *Yd*.
- A set of original (unencoded) examples ($x'_a$, $x'_b$, $\cdots$) NOT present in the original set (*Xd*), with associated labels ($y'_a$, $y'_b$, $\cdots$) obtained by inference from the trained neural network.
- For the scenario in which Party C is part of Party B, Party C may have general knowledge of the type of data sharing project between Party A and Party B.

For the last supposition, the knowledge of the type of data sharing project could imply some understanding of the form of original examples, but it would *not* include actual training or test examples in *Xd*. Knowledge of the form of examples in a given project might represent limited knowledge of what type of images are being used for the project (the attacker might know, for example, that a project concerns computed tomographic (CT) images of the abdomen). In such a scenario, the attacker might know that all unencoded examples are expected to have a surrounding area of black or near-black pixels corresponding to the density of air on a CT, but they would not have data regarding the unencoded data themselves and would not have knowledge of the shape of those data, or the length of unencoded flattened original vectors in *Xd*, prior to the encoding process.

## 3 Description of the encoding methods

The central method of the secure encoding approach presented here is a random non-orthogonal and nonlinear transformation comprised of one or more random non-orthogonal and nonlinear transformations together with one or more random orthogonal transformations. At least one of the included random functions applied to the data is a random orthogonal transformation, which can use a random orthogonal matrix of appropriate size, or can apply an example-wise fixed shuffling (index shuffling) transformation, which represents a subset of the broader set of random orthogonal matrix transformations. For a tensor of rank *N*, sequential transformations

are applied to the input tensor, including at least one step that shuffles each of the examples in a fixed fashion using a stored shuffled index array (or at least one step that dots a random orthogonal matrix of appropriate size with the input vectors). All tensors in a given project's training and test sets, as well as any inference examples, are transformed in similar fashion.

To illustrate the method, three different forms of manipulation, which may be applied in varying combinations, are described below: *padding*, *perturbation*, and *random orthogonal transformation*. While each of these transformations may be applied to tensors of arbitrary rank with any data type suitable for any type of machine learning operation, in this paper, method illustrations and performance experiments will use supervised training of fully-connected artificial neural networks (ANNs) on the MNIST dataset of grayscale ($n$,28,28) handwritten digits (Lecun et al., 1998) as an exemplar, with confirmatory analyses performed on Fashion-MNIST (Xiao et al., 2017) and CIFAR-10 (Krizhevsky).

## 3.1 Padding

*Padding* of a dataset involves generating a random pad of data such that the examples of the source tensor are increased in size (an increase in the flattened vector dimensions of the examples), with resulting examples of uniform shape across the resulting tensor. The random padding may represent one or more rows of data, or one or more columns of data, or a combination of one or more rows and columns (as shown in the L-shaped pad in Figure 2B), or be an addition of any number of pixels with random values in various locations inside or outside of the example images. Different forms of padding may be used that vary in the way they are applied to the input tensor. Three variations on random padding will be described here and performance-tested below: *fixed padding*, *random non-fixed padding*, and *adjusted non-fixed padding*.

### 3.1.1 Fixed padding

With *fixed padding*, a single pad is randomly generated for a source example, typically by random choice for each pixel, within the range of pixel values of the source tensor. This same pad is appended in identical fashion to each example across the $n$ examples of the source tensor, with the same pad used for all examples of the training tensor, the test tensor, and any inference examples.

### 3.1.2 Non-fixed padding

*Non-fixed padding* is any random padding approach that varies from example to example. In one version of non-fixed padding, *random non-fixed padding*, a completely different random pad is generated for each example across the training, test, and inference tensors. In another version of non-fixed padding, *adjusted padding* starts with a fixed pad of appropriate shape that is stored in memory. This stored pad in memory is then adjusted, pixel by pixel, by a random amount in a specified range (by an analogous process to that applied in the perturbation transformation discussed below), and differently adjusted pads are then applied to each example across the training, test, and inference tensors. Other forms of non-fixed padding, not illustrated here, include methods such as generating a group of random pads that is smaller than the number of examples in an input tensor and then using random choice from this pool of random pads to pad the examples in a tensor.

## 3.2 Perturbation

Perturbation of a dataset involves applying a set of random functions that perturb the pixel value at each position in the examples of an input tensor. In the present experiments, the version of this perturbation shown is an alteration of the pixel values by different percentages of the pixel value range.

First, an array of random values from a chosen minimum to a chosen maximum (e.g., min = -5% to max = +5%) is generated, where the array size is matched to the size of an example in the input tensor. (In the experiments shown in this paper, source data are transformed from 8-bit depth [range 0-255] to 32-bit floating-point precision [range 0.0 to 1.0] to enable graded changes between the original 8-bit values.) For each example in the input tensor, the pixel value at a given location is perturbed by the randomly chosen percentage amount at the corresponding position in the perturbation array.

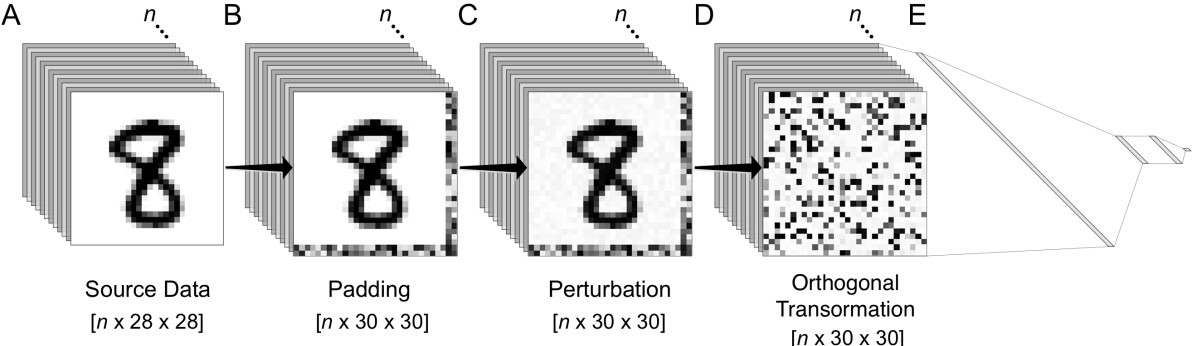

Figure 2: Random *padding*, *perturbation*, and *orthogonal transformation* performed serially prior to training of an artificial neural network (ANN).

A. Illustration of an original dataset ($Xd$), a ($n$,28,28) tensor, the MNIST handwritten digits dataset with $n$ examples, with an illustrated example showing a handwritten digit corresponding to the number 8, with the label category of 8 in the associated label set ($Yd$).

B. Application of a random *padding* transformation. In this case, an L-shaped pad of randomly chosen pixels is generated with 2 rows and 2 columns. This can be *fixed padding* across the examples of the tensor (one randomly generated pad appended in identical fashion to every example), or it can vary across the examples (*non-fixed padding*: a different pad appended to each of the examples). Padding may be any addition of random $x$ rows or $y$ columns or any addition of random extrinsic pixels outside or inside of the examples, as long as the transformed result has a consistent shape. The result in the present example is a transformed tensor of shape ($n$,30,30).

C. Application of a random *perturbation* transformation. A random array of perturbation values (in the illustrated case, from the range of -5% to +5%) is created and used to alter the pixels by a percentage of the original pixel values of the examples of the (n,30,30) tensor. Cases in which pixel values would be taken beyond the minimum or maximum pixel value at a given location can be handled in different ways - in this case, pixels exceeding the minimum or maximum values are clipped to the minimum or maximum value. The effect of the fixed perturbation transformation can be seen in the figure as a fine amount of light gray pixel noise that is most notable when comparing the white pixels in [B] to corresponding positions in [C]. The perturbation transformation may be a *fixed perturbation* (the same set of mathematical perturbations applied to every example) or a *non-fixed perturbation* (different sets of mathematical perturbations applied to each example).

D. Application of a random *orthogonal transformation*. In this case, a fixed index shuffling is applied to shuffle the pixel index positions of each example in identical fashion. More broadly, any random orthogonal transformation may be applied by dotting all examples by a random orthogonal matrix of appropriate size (e.g. for a 30x30 input, a [(30x30) x (30x30)] = [900 x 900] random orthogonal matrix).

E. The random functions applied serially from [B] through [D] are independent of the associated labels in $Yd$ and produce an aggregate random non-orthogonal and non-linear transformation, and the result of this transformation is used to train a multilayer ANN.

As pixel values are randomly perturbed upward or downward by this technique, perturbation beyond the relevant minimum and maximum values must be explicitly handled. While many approaches can be used, in this paper, two approaches are tested: *clipping* and *reflection*. Clipping assigns the minimum or maximum value to any pixel values brought beyond these values by the perturbation algorithm. For example, if a pixel value on a float scale from 0.0 to 1.0 started at 0.99 and would be brought to a value of 1.03 by a perturbation, clipping changes the resulting value to 1.0. Reflection brings the value to a corresponding amount within the min/max boundaries. For example, if a pixel value on a float scale from 0.0 to 1.0 started at 0.99 and would be brought to 1.03 by a perturbation, the amount exceeding the maximum value is (1.03 - 1.0 = 0.03) and the assigned value using reflection would be (1.0 - 0.03 = 0.97). Figures 2C and 3B illustrate the effect of applying a -5% to +5% bounded percentage perturbation with clipping to the examples of an input tensor. With very large perturbation settings, it is possible for reflection to occur back and forth between the max and min values. Throughout this paper, the default setting for handling perturbations beyond the maximum and minimum bounds will be clipping unless stated otherwise.

Different forms of perturbation may be applied that vary in their application to the input tensor. Two variations on perturbation will be described here and performance-tested: *fixed perturbation*, in which all examples are modified in similar fashion, and *non-fixed perturbation*, in which each example is modified in a different fashion.

### 3.2.1 Fixed Perturbation

In *fixed perturbation*, a single random perturbation array is generated, and the set of pixel value perturbations using the perturbation array is applied in identical fashion to each of the examples in the input tensor, with the same fixed perturbation transformation applied to all of the examples of the training tensor and the test tensor, as well as any inference examples.

### 3.2.2 Non-fixed perturbation

In *non-fixed perturbation*, a different random perturbation array is generated and applied to each example. For example, a different set of pixel variation functions may be generated for each example with a minimum to maximum range for the degree of potential perturbation (e.g. min = -5% to max = +5% perturbation of pixel values). With non-fixed perturbation, different perturbation transformation arrays are applied to each of the examples of the training tensor and test tensor, as well as to any inference examples.

## 3.3 Orthogonal Transformation

Application of an orthogonal matrix transformation of input vectors has been proposed as a method to encode data for use with artificial neural networks (Chen, 2019), because orthogonal matrix transformation maintains vector length and angles between vectors and thus machine learning accuracy should be maintained by this approach. In the proposed method of Chen (2019), a square image file of appropriate size is used as a key that is subjected to QR factorization to yield an orthogonal matrix. The orthogonal matrix thus obtained is used to encode training and test data. In the methods presented here, the orthogonal matrix transformations applied are either drawn randomly from the O(N) Haar distribution or applied by fixed shuffling (index shuffling) of pixel values in each example and used in combination with steps of random padding and random perturbation to yield an aggregate non-orthogonal and nonlinear transformation.

Two subtypes of orthogonal transformation are used here in combination with the transformations of padding and perturbation described above. In the broadest method, a *random orthogonal matrix* of appropriate size is drawn randomly from the O(N) Haar distribution and used to dot input vectors. In a specific subtype of random orthogonal matrix transformation, *fixed shuffling* (index shuffling) is applied to the pixels of input vectors.

### 3.3.1 Random orthogonal matrix transformation

To transform input vectors by random orthogonal transformation, a random orthogonal matrix is generated, either by random choice of an orthogonal matrix of appropriate shape drawn from the O(N) Haar distribution,

or, alternatively, by initial generation of a random matrix of appropriate shape that is then subjected to QR factorization to obtain a random orthogonal matrix of appropriate shape. Once a random orthogonal matrix is generated by either method, it is stored in memory and used to transform input vectors by taking the vector · matrix dot product of each input vector and the stored random orthogonal matrix. The same random orthogonal matrix is used to transform training, test, and inference data in the same fashion.

### 3.3.2 Fixed shuffling

*Fixed shuffling*, or index shuffling, of a dataset involves shuffling the pixel index locations of the examples of an input tensor such that the same shuffling rearrangement is applied in identical fashion to each example in the tensor. Fixed shuffling is a subset of the broader set of random orthogonal matrix transformations. Fixed shuffling can be accomplished by randomly generating a shuffling index array that matches the shape of an example in the input tensor, which stores the instructions for rearranging the pixel locations in each example. For example, a shuffling index array might indicate that the pixel residing at location (0,0) in each example be repositioned to position (16,9) in each example in the fixed shuffled tensor, with a corresponding repositioning of every pixel index location in each example. The identical result may, of course, be obtained by flattening the examples, using a flat shuffling index array of corresponding length, then reshaping the result back to the original $(x,y)$ example shape, and fixed shuffling may be applied to examples of any dimension in tensors of any rank. Alternatively, fixed shuffling may be applied by generating a square orthogonal matrix filled with zeroes except for single ones in unique column positions across the matrix rows, and this matrix may be used to have the same index shuffling (coordinate axis permutation) effect. The same fixed shuffling transformation is applied to the examples of the training tensor and the test tensor, as well as any inference examples. Figure 2D shows the effect of applying a fixed shuffling transformation to the examples of an input tensor, in this case the result of the random padding transformation shown in Figure 2B followed by the random perturbation transformation shown in Figure 2C.

### 3.3.3 Non-fixed shuffling

*Non-fixed shuffling* (in which a different shuffle is applied to each example of the training and test datasets) is of limited practical utility in the context of training machine learning systems for inference—separate example-wise shuffling disrupts the spatial relationships of the data elements across examples, thus requiring a machine learning system to train on non-spatial data alone. This effect is, however, of academic interest, because it allows for a quantitative assessment of the extent to which a particular machine learning system uses spatial vs. non-spatial information to train. Specifically, training on a fixed shuffled dataset involves both spatial information (e.g., the arrangement of pixels in the original dataset) and non-spatial information (e.g., the percentage of darker pixels in some examples compared with others, which can in turn associate with the example label), while training on a non-fixed shuffled dataset involves only non-spatial information. A comparison between the results of these two treatments (fixed shuffling vs. non-fixed shuffling) thus allows for a quantitative assessment of how these two aspects of a given dataset (spatial vs non-spatial information) contribute to network training.

### 3.4 Serial combinations of transformations

As shown in Figure 2, the transformations of padding, perturbation, and orthogonal transformation may be performed sequentially, in varying combinations. (The sequential application of multiple transformations can, of course, be established as a single algorithmic transformation that has the same effect). At least one random orthogonal transformation step, as described above, is used, and typically an orthogonal transformation step is deployed at the final stage of the chain of transformations, as illustrated in Figure 2. Multiple rounds of random functions may be applied, including multiple versions of each category of transformation; e.g., both random orthogonal matrix transformation and fixed shuffling may be sequentially applied. The effect of combining the non-orthogonal and nonlinear transformation steps of padding, perturbation, or a combination of these, is to create an aggregate random non-orthogonal and nonlinear transformation.

## 4    Methods

### 4.1    Experimental setup

Training performance of the encoding steps of padding, perturbation, and orthogonal transformation, alone and in different combinations, was tested here by training and validation with the MNIST dataset (Lecun et al., 1998), with the usual 60K training and 10K validation split. Training and validation were performed using the same fully-connected ANN structure and hyperparameters for all trained models, using TensorFlow / Keras in Python 3.6 on Google Colab+, running on a NVIDIA Tesla T4 or P100 GPU, depending on the server connection. All training sessions for transformed datasets and paired controls were performed on the same server connection, using the same available GPU. For the training time comparisons described below, a server connection running a NVIDIA Tesla T4 GPU was used. ANN structure was (1) an input layer corresponding to the example data shape with or without change in shape produced by encoding, (2) two fully-connected hidden layers of 128 neurons each with ReLU activation, with 0.25 Dropout, Flatten, and 0.5 Dropout, and (3) an output layer of 10 neurons corresponding to the 0-9 digit labels of MNIST, with Softmax activation. The loss function was categorical crossentropy, and the optimizer was RMSprop, with a learning rate of 0.001, rho of 0.9, and epsilon of 1e-07. Training batch size was 128, with 12 training epochs for all training runs. To facilitate appropriate comparisons of validation accuracy and of training time, model parameters were fixed for all experiments and not individually optimized—the only difference from model to model was any required increase in input shape produced by the encoding process (specifically the increase in input shape produced by padding). To achieve appropriate precision of fixed perturbation transformations, all pixel values for all models were converted to floating point precision.

Each encoding process that was tested, whether individual encoding steps or multiple encoding steps in different combinations, was performed a total of 15 separate times (see *sample size calculations* below), with separate fully-connected ANN models trained for each from scratch. For example, to test orthogonal transformation alone, as shown in Figure 3C, a new random orthogonal matrix was generated to transform the training and validation splits of MNIST, and then the model underwent training. This was repeated 14 more times, each time deleting all model variables, generating a new random orthogonal matrix, transforming the tensors *de novo*, and training a new model from scratch. During the same session, while connected to the same Google Colab+ cloud server with the same available GPU, control models were independently trained using MNIST without any encoding steps. For each comparison, 15 control models were trained, with presented controls paired to the same server session used for the encoding process being tested, to ensure similar conditions (such as available GPU) and to avoid multiple comparisons in statistical analysis.

Additional analyses were performed with similar network structures using additional datasets (Fashion-MNIST (Xiao et al., 2017), CIFAR-10 (Krizhevsky)), with minor modifications to network structure as described in the Results.

### 4.2    Statistics and sample size calculation

For comparison of validation accuracy between controls and a given encoding process, the nonparametric Kruskal-Wallis equality-of-populations rank test was performed, with P value for the Chi-square test without ties reported for each comparison (similar results were obtained with the parametric Student's t-test). For histogram analysis of pixel value distribution in original and encoded datasets, examples were flattened, with floating-point precision, scaled to 0.0 to 1.0, and histograms with 100 bins each were generated with numpy.histogram in Python and fitting of encoded histogram data to the Gaussian distribution was performed in GraphPad 8.4 (GraphPad Software, San Diego, CA), with goodness-of-fit reported as $R^2$. For measurement of information entropy in original and encoded examples, examples were flattened, with floating-point precision, scaled to 0.0 to 1.0, and flattened and scaled signals were converted to probabilities summing to 1.0, then total entropy across each example was determined in *nats* ($1/\ln(2)$ *shannons*) by S = - sum(pk * ln(pk). Linear regression of original and encoded entropy for pairs of examples within label groups was performed in GraphPad with goodness-of-fit reported as $R^2$.

Sample size power analysis was performed for two-sample means using a series of 20 control runs (unencoded MNIST) on the same ANN, which showed a control mean validation accuracy of 0.977 and standard devi-

ation of the validation accuracy of 0.001, yielding a very small sample size because of the extremely tight distribution of results reflected in the very small standard deviation. Conservatively assuming a standard deviation 50x larger than this (0.05) for both groups yields a sample size estimation of 10 model trainings needed per group (control vs. encoded) to detect a difference in accuracy of 0.007 (0.977 for control, 0.970 for encoded) with power of 0.8 and alpha of 0.05. Keeping the very low observed standard deviation for the control group (0.001) but conservatively assuming a 50x higher standard deviation for the encoded group (0.05) for the same detection of a 0.007 difference in accuracy yields a sample size estimation of 7 model trainings per group. Based on the above calculations, a conservative choice of 15 training runs per group was used for experiments comparing performance between control and encoded data, and for training time experiments, 20 training runs per group was used. For training time experiments, elapsed time was measured from the start of the first training epoch to the end of the last training epoch.

Statistical tests and sample size calculations were performed with Stata/MP 16.1 (StataCorp, College Station, TX) and GraphPad. Illustrations and graphs were generated using the Python libraries Matplotlib and Toyplot in Google Colab+ and using GraphPad.

## 5 Results

### 5.1 Single transformations

Figure 3 shows model training performance for each of the random functions described above (padding, perturbation, and orthogonal transformation) applied as separate, single-step transformations and compared with training with unencoded MNIST (control). For padding transformation (Figure 3A$_{1\text{-}3}$), the results of fixed padding (application of the same random pad to every example) are shown in Figure 3A$_2$, and the results of non-fixed padding (application of a different random pad to every example) are shown in Figure 3A$_3$. For perturbation transformation (Figure 3B$_{1\text{-}3}$), the results of fixed perturbation (application of the same set of perturbation transformations to every example) are shown in Figure 3B$_2$, and the results of non-fixed perturbation (application of a different set of perturbation transformations to every example) are shown in Figure 3B$_3$. For orthogonal transformation (Figure 3C$_{1\text{-}3}$), results of random orthogonal matrix transformation are shown in Figure 3C$_2$, and the results of fixed shuffling are shown in Figure 3C$_3$. For each of the transformations applied individually, no significant difference in the validation accuracy was detected between models trained on transformed data vs. control (Figure 3).

#### 5.1.1 Non-fixed vs. fixed shuffling

As discussed above, non-fixed shuffling (application of a different index shuffle to each example across the training and test tensors) is not generally of practical use, as this manipulation disrupts the spatial relationships between data elements (pixels) and thus forces the network to train only on non-spatial information. This effect does, however, allow for an assessment of the extent to which a machine learning system uses spatial vs. non-spatial information to train. Comparing network training between non-fixed shuffled and fixed shuffled MNIST showed that the non-fixed shuffled data (lacking spatial information) yielded a mean validation accuracy of $0.2608 \pm 0.0049$ compared with $0.9769 \pm 0.001$ for fixed shuffled controls (P<0.001).

### 5.2 Variation in padding size

The size of the appended random pad is one of several adjustable parameters in the methods tested here, so the impact of varying the size of the appended pad on validation accuracy was tested for both fixed and non-fixed padding. Figure 4 shows the results of increasing pad size from 0 rows (control) to 10 rows of padding, with examples illustrated in Figure 4A. For both fixed padding and non-fixed padding, increasing size of the appended pad did not reduce validation accuracy (Figure 4B).

### 5.3 Variation in perturbation range

Another adjustable parameter in the present methods is the range of potential perturbation of pixels in the examples. In the above illustrations (Figures 2C and 3B1), a perturbation range of -5% to +5% is shown,

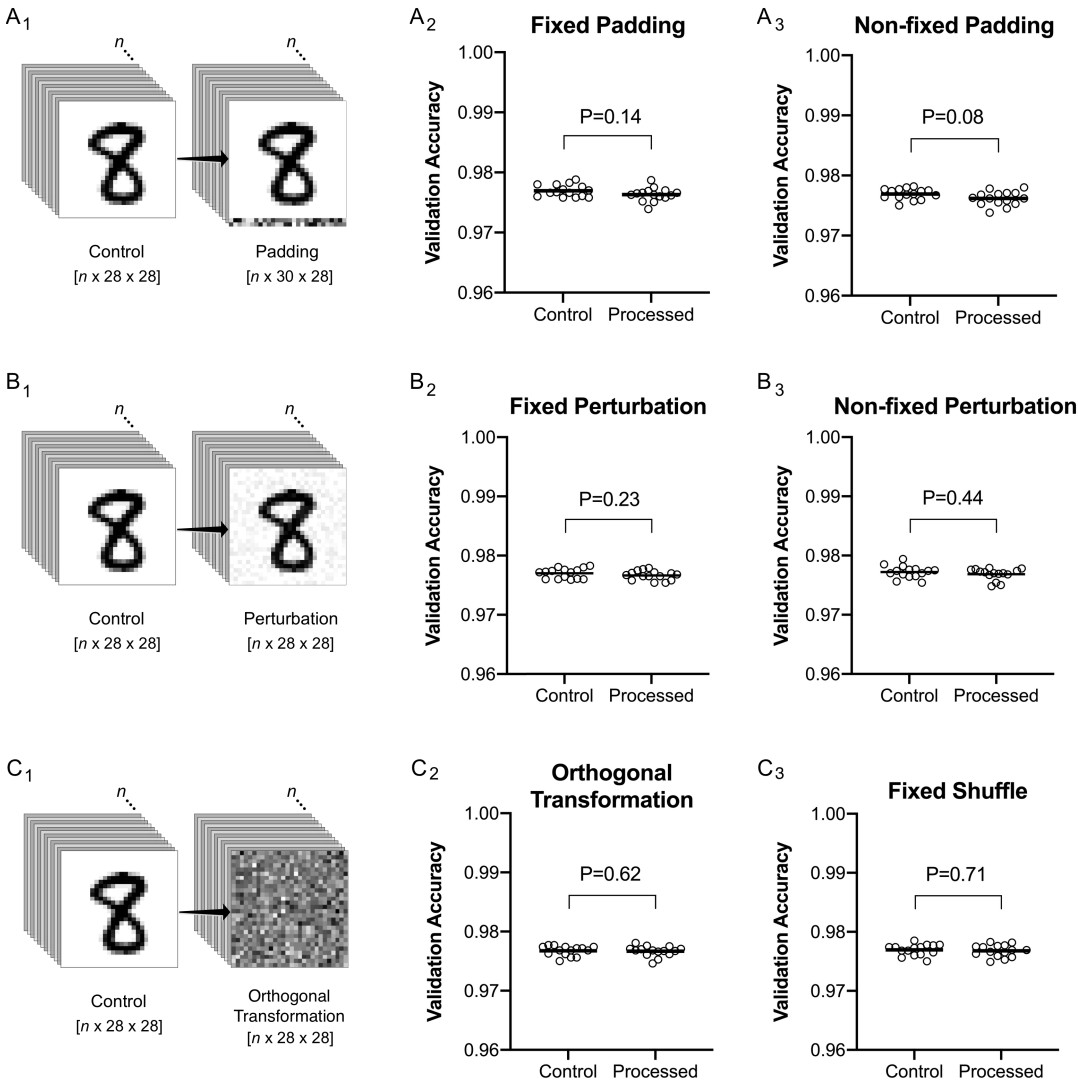

Figure 3: Separate transformations compared with controls.

$A_1$. Illustration of random *padding* (a 2-row random pad appended to the MNIST examples).

$A_2$. Model training on *fixed padded* data (same random pad for every example). Mean validation accuracy was $0.9763 \pm 0.001$ for padded data (Processed), and $0.9770 \pm 0.001$ for control (P=0.14).

$A_3$. Model training on *non-fixed padded* data (a different random pad for each example). Mean validation accuracy was $0.9762 \pm 0.0012$ compared with $0.9769 \pm 0.0009$ for control (P=0.08).

$B_1$. Illustration of random *perturbation* (applying a random perturbation array, constrained from -5% to +5% to the examples of MNIST).

$B_2$. Model training after *fixed perturbation* (same random perturbation array applied to every example). Mean validation accuracy was $0.9767 \pm 0.0008$ (Processed) compared with $0.9770 \pm 0.0009$ (Control) (P=0.23).

$B_3$. Model training after *non-fixed perturbation* (a different array of random perturbations applied to each example). Mean validation accuracy was $0.9768 \pm 0.001$ (Processed) compared with $0.9772 \pm 0.001$ (Control) (P=0.44).

$C_1$. Illustration of *random orthogonal transformation.* A single random orthogonal matrix is used to transform every example.

$C_2$. Model training after *random orthogonal matrix transformation* (as illustrated in $C_1$). Mean validation accuracy was $0.9767 \pm 0.0009$ (Processed) compared with $0.9767 \pm 0.0008$ (Control) (P=0.62).

$C_3$. Model training after *fixed shuffling* (a subset of the broader set of random orthogonal transformations). Mean validation accuracy was $0.9768 \pm 0.001$ (Processed) compared with $0.9769 \pm 0.0009$ (Control) (P=0.71).

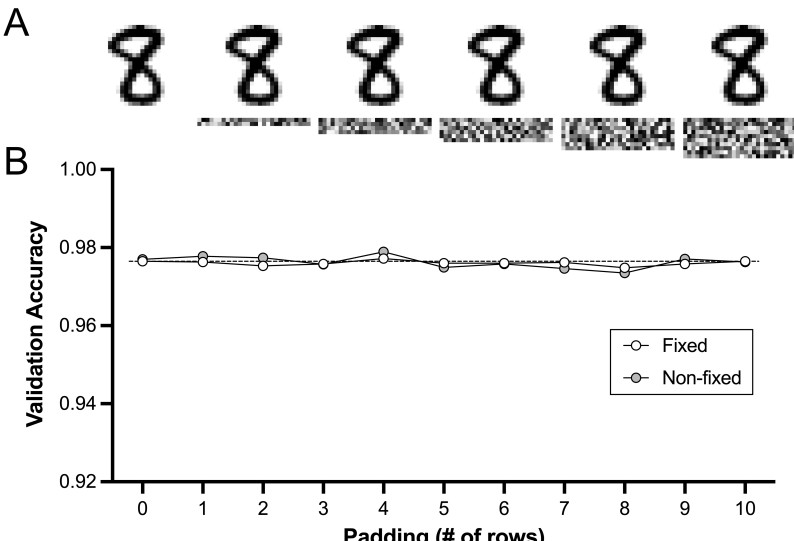

Figure 4: Variation of padding size.
A. Illustration of increasing pad size from 0 pad rows on the left (control) up to 10 pad rows on the right.
B. Validation accuracy of models trained with a single-step transformation using different numbers of pad rows, from 0 to 10 pad rows. White circles show the results of varying pad size using *fixed padding* (the same random pad applied to every example) and gray circles show the results of varying pad size using *non-fixed padding* (a different random pad applied to each example). Horizontal dotted line indicates the baseline from control.

but this range is arbitrary and can be varied across smaller or larger ranges, creating different degrees of distortion for human viewers depending on the range choice. While only small ranges of perturbation would typically be desirable to achieve security goals together with padding and orthogonal transformation, it is important to establish how varying degrees of perturbation impact model training.

As discussed above, pixel perturbation beyond the bounds of the available pixel values may be handled in several different ways, including clipping (setting the value to the boundary value) and reflection (reflecting the value about the boundary value), and, as expected, these two different approaches yield visibly different results when very large perturbation ranges are used, as seen in Figure 5A (clipping) vs. Figure 5C (reflection). The impact of varying the available perturbation range on validation accuracy is shown in Figure 5B (clipping) and Figure 5D (reflection).

Very large perturbation ranges with clipping cause an increasing number of pixel locations to be clipped, and as expected, this leads to a progressive decrease in accuracy, but to a lesser extent than might be predicted based on visual inspection of the encoded results. As shown in Figure 5B, perturbation ranges up to -50% to +50% do not appreciable decrease validation accuracy, and even very large perturbation ranges above -150% to +150% that render the examples uninterpretable to human inspection still produce validation accuracies around 0.95 with MNIST.

Very large perturbation ranges with reflection are expected to retain more useful training information than clipping, as clipping is intrinsically a lossy manipulation, while reflection need not be. Consistent with this, increasing the perturbation range with reflection causes a smaller decrease in validation accuracy than that seen with clipping, reaching a plateau of validation accuracy around 0.965 (Figure 5D).

## 5.4 Serial application of padding, perturbation, and orthogonal transformation

The present method involves sequential application of random padding, random perturbation, and random orthogonal transformation to yield an aggregate non-orthogonal and nonlinear transformation. While the

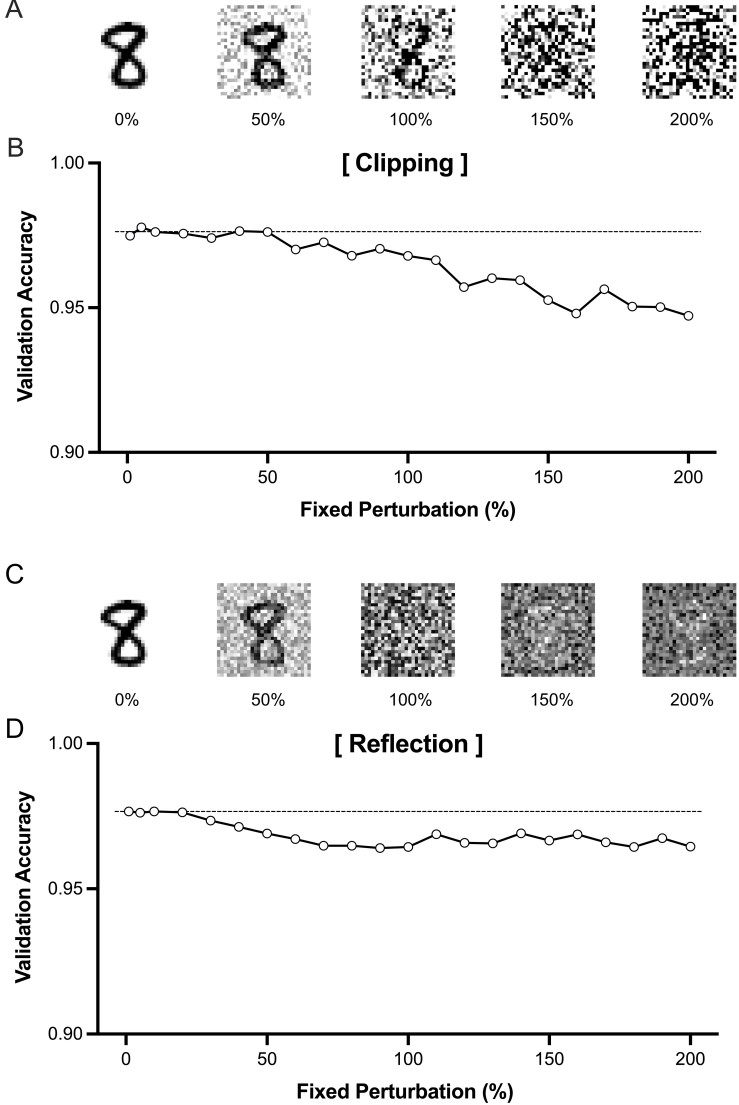

Figure 5: Variation of perturbation range.
A. Illustration of perturbation ranges from 0% (control) to 200% (-200% to +200%), processed with *clipping* to handle values beyond the pixel value range.
B. Validation accuracy of models trained with a single-step transformation using fixed perturbation with clipping, from 0% to 200%. Horizontal dotted line indicates the control baseline.
C. Illustration of perturbation ranges from 0% to 200%, processed with *reflection* to handle values beyond the pixel value range.
D. Validation accuracy of models trained with a single-step transformation using fixed perturbation with reflection, from 0% to 200%. Horizontal dotted line indicates the control baseline.

results presented in Figure 3 show that single transformations of padding, perturbation, or orthogonal transformation do not decrease validation accuracy, it is critical to test the impact of different forms of padding and perturbation performed in series with orthogonal transformation, in a typical sequence of (padding → perturbation → orthogonal transformation [using the fixed shuffling form of orthogonal transformation in the examples shown]).

### 5.4.1 Different padding approaches combined with fixed perturbation and orthogonal transformation

The different forms of random padding described above (fixed padding, adjusted padding, and non-fixed padding) may be applied together with fixed perturbation and orthogonal transformation, so each of these approaches was tested to train models on data encoded with the sequence (padding → fixed perturbation → orthogonal transformation).

As shown in Figure 6A, application of fixed padding together with fixed perturbation and fixed shuffling produces a very small but significant decrease in validation accuracy (0.9754 for the encoded data compared with 0.9769 for control, a relative decrease in validation accuracy of 0.15%). Adjusted padding (see above for detailed description) using a potential adjustment of -20% to +20% together with fixed perturbation and fixed shuffling yielded a similar validation accuracy (0.975). Application of non-fixed padding (a different random pad for every training and test example) together with fixed perturbation and fixed shuffling yielded a slightly lower validation accuracy (0.9742 for the encoded data, a relative decrease in validation accuracy of 0.29%).

Similar results were obtained when random orthogonal matrix transformation was used instead of fixed shuffling in the above combinations of padding and fixed perturbation (data not shown).

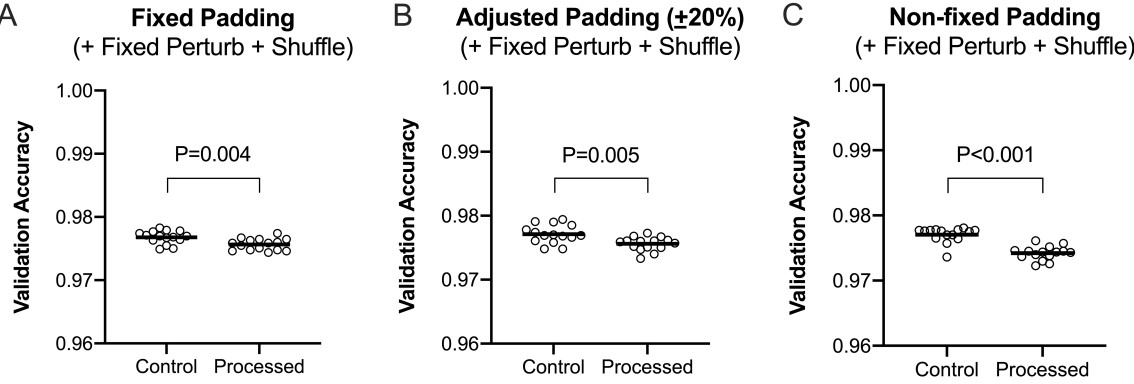

Figure 6: Different padding approaches combined with fixed perturbation and fixed shuffling.
A. *Fixed padding* (the same 2x28 pad appended to every example), followed by fixed perturbation (-5% to +5%), followed by fixed shuffling. Mean validation accuracy was $0.9754 \pm 0.001$ (Processed) compared with $0.9769 \pm 0.0011$ (Control) (P=0.004).
B. *Adjusted padding* (a 2x28 pad that is randomly perturbed within the bounds [-20% to +20%] prior to appending to each example), followed by fixed perturbation of the examples (-5% to +5%), followed by fixed shuffling. Mean validation accuracy was $0.9756 \pm 0.0011$ (Processed) compared with $0.9771 \pm 0.0014$ (Control) (P=0.005).
C. *Non-fixed padding* (a different 2x28 random pad appended to each example), followed by fixed perturbation (-5% to +5%), followed by fixed shuffling. Mean validation accuracy was $0.9742 \pm 0.001$ (Processed) compared with $0.9770 \pm 0.0012$ (Control) (P<0.001).

### 5.4.2 Different padding approaches combined with non-fixed perturbation and orthogonal transformation

The different forms of random padding (fixed padding, adjusted padding, and non-fixed padding) may also be applied together with non-fixed perturbation and orthogonal transformation, so each of these approaches was tested to train models on data encoded with the sequence (padding → non-fixed perturbation → orthogonal transformation [using the fixed shuffling form of orthogonal transformation in the examples shown]).

As shown in Figure 7A, application of fixed padding together with non-fixed perturbation and fixed shuffling produces a small but significant decrease in validation accuracy (0.9750, a relative decrease in validation accuracy of 0.23%). Adjusted padding using a potential adjustment of -20% to +20% together with non-fixed perturbation and fixed shuffling yielded a similar validation accuracy (0.9750, a relative decrease in validation accuracy of 0.21%). Application of non-fixed padding together with non-fixed perturbation and fixed shuffling yielded a lower validation accuracy (0.9738 for the encoded data, a relative decrease in validation accuracy of 0.35%).

Similar results were obtained when random orthogonal matrix transformation was used instead of fixed shuffling in the above combinations of padding and non-fixed perturbation (data not shown).

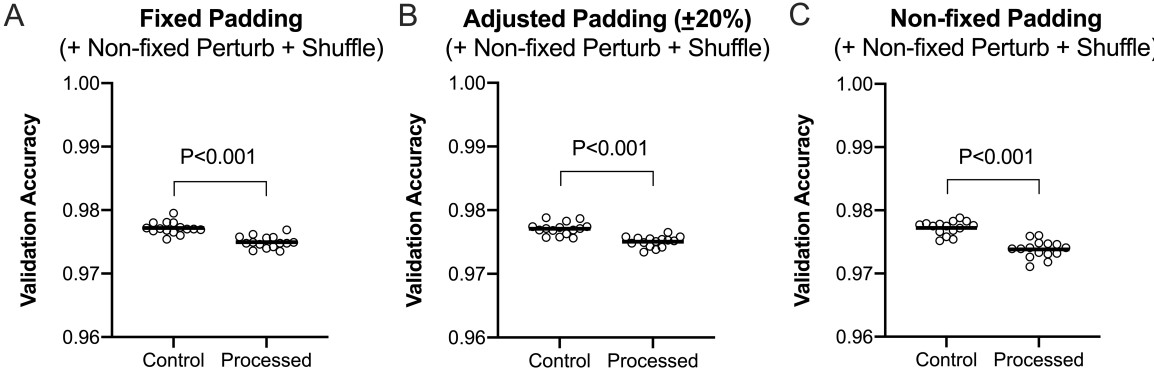

Figure 7: Different padding approaches combined with non-fixed perturbation and fixed shuffling.
A. *Fixed padding*, followed by *non-fixed perturbation* (-5% to +5%), followed by fixed shuffling. Mean validation accuracy was $0.9750 \pm 0.001$ (Processed) compared with $0.9772 \pm 0.001$ (Control) (P<0.001).
B. *Adjusted padding*, followed by *non-fixed perturbation* of the examples (-5% to +5%), followed by fixed shuffling. Mean validation accuracy was $0.9750 \pm 0.0008$ (Processed) compared with $0.9771 \pm 0.0010$ (Control) (P<0.001).
C. *Non-fixed padding*, followed by *non-fixed perturbation* (-5% to +5%), followed by fixed shuffling. Mean validation accuracy was $0.9738 \pm 0.0014$ (Processed) compared with $0.9772 \pm 0.0011$ (Control) (P<0.001).

### 5.5 Training time

Training times were tested to compare computation time for control unencoded MNIST with the same model specification trained using MNIST encoded with sequential padding, perturbation, and fixed shuffling.

Model training with MNIST encoded with fixed padding followed by fixed perturbation followed by fixed shuffling showed no significant difference in training times when compared with control MNIST ($25.92 \pm 0.57$ sec for transformed MNIST vs. $26.06 \pm 0.65$ sec for control, Figure 8). The mean processing time required to perform encoding with these three transformations across the training and test tensors (representing a one-time processing time for a given project) was $22.49 \pm 0.23$ sec, which represents about 0.3 msec per MNIST example. Of note, the code used to perform these serial transformations was not optimized to minimize the required processing time for encoding, as the brief processing times required here were acceptable for the present work.

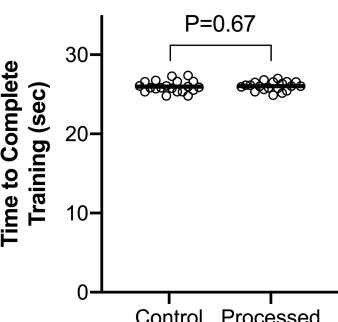

Figure 8: Model training time.
Model training for a fixed padded (2x28), fixed perturbed (-5% to +5%), and fixed shuffled data set (Processed) was $25.92\pm0.57$ sec, compared with $26.06\pm0.65$ sec for control (P=0.67). For this set of transformations (fixed padding, fixed perturbation, and fixed shuffling), processing of MNIST prior to model training took $22.49\pm0.23$ sec, which represents a one-time processing step prior to training for a given project. Processing times were tested on a Google Colab+ server running on a NVIDIA Tesla T4 GPU, with 20 separate model trainings per condition.

## 5.6   Testing on alternative data sources

To explore whether similar results may be obtained with datasets known to be more challenging than MNIST, additional experiments tested fully-connected ANN training on encoded and control versions of Fashion-MNIST (Xiao et al., 2017) and CIFAR-10 (Krizhevsky). Encoding used on the additional datasets was 2-row non-fixed padding, non-fixed perturbation (-5% to +5%), and fixed shuffling. Training on the Fashion-MNIST database yielded the following validation accuracies: control $0.8999\pm0.0032$, encoded $0.8874\pm0.0025$ (P<0.001). Training on CIFAR-10 yielded: control $0.5505\pm0.0174$, encoded $0.5289\pm0.033$ (P<0.001). For Fashion-MNIST and CIFAR-10, model structures were similar to those used for MNIST, but with 256 batch size, 24 epochs, and 256 neurons in the first layer. The typical approach to obtain better accuracies with these datasets, particularly with CIFAR-10, involves the use of CNN structures, which will be possible to integrate with our encoding method in the future (see below), but are beyond the scope of this initial report. Finally, our synthetic challenge dataset (58x58 pixel images as encoded, 50K train, 10K test, see below for details) yielded validation accuracies of: control $0.9849\pm0.0016$, encoded $0.9726\pm0.0028$, P<0.001. For the models trained on the challenge dataset, the structures were similar to those used for MNIST, but with 682 neurons in both layers. The absolute decrease in validation accuracy for each dataset from control to encoded version is shown in Table 1.

Table 1: Absolute decrease in validation accuracy across datasets

| Dataset | Percent accuracy decrease |
|---|---|
| MNIST | 0.51 |
| Fashion-MNIST | 1.25 |
| CIFAR-10 | 2.16 |
| Challenge Dataset | 1.23 |

## 5.7   Privacy Analysis

As described above, our threat model has a potential attacker in possession of an encoded training dataset ($Zd$) and $n$ corresponding labels ($Yd$); a trained neural network, obtained by training on $Zd$ and $Yd$; and a

set of attacker-provided original (unencoded) examples ($x'_a$, $x'_b$, $\cdots$) *not* present in the original dataset ($Xd$), with associated labels ($y'_a$, $y'_b$, $\cdots$) obtained by inference from the trained neural network. Our privacy scheme is designed to protect against an attacker using these elements to reverse our encoding random function $f(Xd) \rightarrow Zd$ by discovering the inverse function $f^{-1}(Zd) \rightarrow Xd$, or a close approximation thereof.

The random functions that create an aggregate non-orthogonal and nonlinear transformation in our method produce an extraordinarily large search space in which a potential attacker would have to determine the encoding transformation. For example, index shuffling of a given array yields a single result at random out of a very large set of possible shuffled arrays, with the number of possible arrays represented by the factorial of the length of the array. Thus, even for small example arrays such as those in MNIST, with (28x28 = 784 pixel) examples, the total number of possible index shuffles for an example is 784!, which is $3.19 \times 10^{1930}$ possible index shuffles, a number vastly larger than the number of guesses required to brute-force AES-256 ($2^{256}$, or roughly $10^{77}$). Beyond the particular case of index shuffling, the space of all possible random orthogonal matrix transformations for a given input vector size is far greater.

Random padding, either fixed or non-fixed, changes the number of vector dimensions between $Xd$ and $Zd$, in a random fashion that is unknown to Party C (the attacker) in the threat model. As shown above in Figure 4, an arbitrary number of vector dimensions may be added to $Xd$, and the location, number, and random pixel values of these new dimensions in the resulting vectors in $Zd$ are all unknown to the attacker. The values stored in the added dimensions are randomly determined for all forms of padding, and for non-fixed padding or adjusted padding, the values stored in the added dimensions randomly vary from example to example. Because a large number of added dimensions with random values may be added non-deterministically between $Xd$ and $Zd$, this creates a 'needle in a haystack' problem: for encoded vectors in $Zd$ of known length, len(z), the attacker must find the original vectors of unknown length, len(x) « len(z). The padding component of the encoding process, which can add random pixels at any position inside or outside of the original input examples, introduces another source of randomness with a very large search space: for the addition of (10 x 28) random pixels to an original 28 x 28 MNIST example, this can be approximated by [ (784+280) combinations 784 ] = $5.8 \times 10^{264}$. When padding is put together with an index shuffle, the search space in which an attacker must discover the original vector in the random padded vector can be approximated by [ (784+280) permutations 784 ] = $1.8 \times 10^{2195}$. Both of these calculations substantially underestimate the the task, given that in our threat model the attacker does not have knowledge of the original number of vector dimensions.

Perturbation adds random extrinsic noise across example vectors that further serves to impede attempts to reduce the search space. The impact of perturbation on the transition of examples from $Xd$ and $Zd$ is not a deterministic function between $Xd$ and $Zd$: every pixel is varied randomly in different fashion, across a range of perturbation values that is unknown to the attacker. Non-fixed perturbation ensures that even before orthogonal transformation is applied, no pixel value in the original examples $Xd$ maps directly to a corresponding pixel value in the encoded examples $Zd$.

Finally, all of the random functions discussed above (padding, perturbation, and orthogonal transformation) are *fully independent of the label categories* in $Yd$—in the next section, this will be examined in detail in the context of our threat model.

## 5.8 Information Leakage

The information leakage in our threat model as shown in Figure 1 is the specific knowledge of the relationship between encoded examples in $Zd$ and labels in $Yd$ and between non-encoded inference examples $x'$ and inference labels $y'$ generated by a model trained on encoded examples (Figure 1).

Because all of the transformations applied in the present method are random functions that are fully independent of the label categories in $Yd$, it can be shown analytically that the information leaked in the threat model does not assist in an attempt to reverse the transformation of $Xd$ to $Zd$.

As described in detail above, the design of the encoding process presented here involves serial application of random functions to the data, beginning with $Xd$ and yielding $Zd$, in a fashion that is independent of the

label category in $Yd$ (Figure 9). Put another way, all of the random functions used are applied across the tensors and are agnostic to label value.

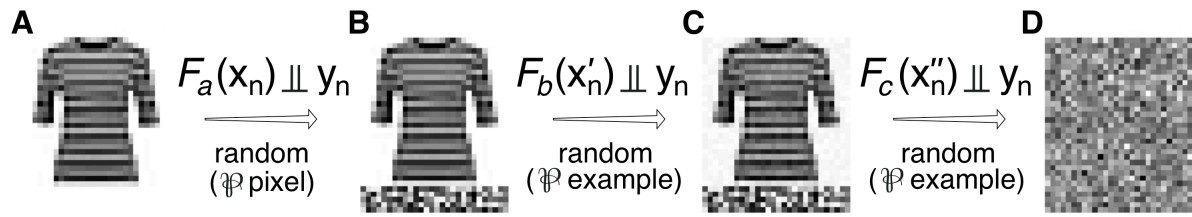

Figure 9: Independence of random function transformations from label categories.
Independence of random transformations from label category, illustrated with a transformation of non-fixed padding, fixed perturbation, and random orthogonal matrix transformation applied to an example $x_n$ from Fashion-MNIST.
A. A random function $F_a$ acting on examples $x_n$ in $Xd$ applies a non-fixed padding to each example $x_n$ to yield a transformed example $x'_n$, where the function is random per pixel of each of the examples. The random function $F_a$ is independent of label $y_n$.
B. A random function $F_b$ acting on examples $x'_n$ applies a fixed perturbation of each example $x'_n$ to yield transformed examples $x''_n$, where the function is random per example across the tensor. The random function $F_b$ is independent of label $y_n$.
C. A random function Fc acting on examples $x''_n$ applies a random orthogonal matrix transformation to each example $x''_n$ to yield a transformed example $z_n$ in the resulting transformed dataset $Zd$, where the function is random per example in the dataset. The random function $F_c$ is independent of label $y_n$.

Because all random functions in the present method are independent of the value of labels in $Yd$ (Figure 9), the information leakage in our threat model (knowledge of the relationship between $Zd$ and $Yd$ and between inference examples $x'$ and inference labels $y'$) does not provide information on the nature of the random functions applied between $Xd$ and $Zd$.

To test this assertion empirically, original images from Fashion-MNIST were transformed by padding (5 x 28 non-fixed padding, random at the pixel level for all examples), fixed perturbation (min = -5% to max = +5%, with reflection), and random orthogonal matrix transformation (as illustrated in Figure 9), and the original and encoded examples were subjected to analysis according to label category.

As shown in Figure 10, the average distribution of pixel values in the original Fashion-MNIST ($Xd$) differ from label to label in $Yd$, as would be expected for images of different types of objects (e.g., T-shirts [label 0] vs. Sandals [label 5]). When encoding by the sequence of non-fixed padding, fixed perturbation, and random orthogonal matrix transformation (as illustrated in Figure 9) is applied to yield an encoded dataset ($Zd$), the distribution of pixel values within label categories in $Yd$ is Gaussian (with $R^2$ goodness-of-fit to the Gaussian distribution >0.997 for all label categories) and does not vary from label to label. This observed lack of variation across label categories is expected from the application of random functions that transform $Xd$ into $Zd$ in a fashion that is independent of labels in $Yd$.

To further show that variability in $Xd$ across label categories in $Yd$ does not carry over to variability in $Zd$ across label categories in $Yd$, information entropy was measured for pairs of original ($Xd$) and encoded ($Zd$) examples, grouped according to label category ($Yd$). As shown in Figure 11, entropy varies considerably across label categories for examples in $Xd$ (Figure 11A), and after encoding by the serial random functions that are independent of $Yd$, entropy rises to a higher, and far more narrowly distributed, level (Figure 11$B_1$ and 11$B_2$). Linear regression between entropy of original examples and paired encoded examples (before and after transformation by random functions independent of $Yd$) shows no relationship between variation in entropy before encoding and the variation in entropy after encoding within label categories in $Yd$ (Figure 11C). This observed lack of a relationship between entropy variation before encoding and entropy

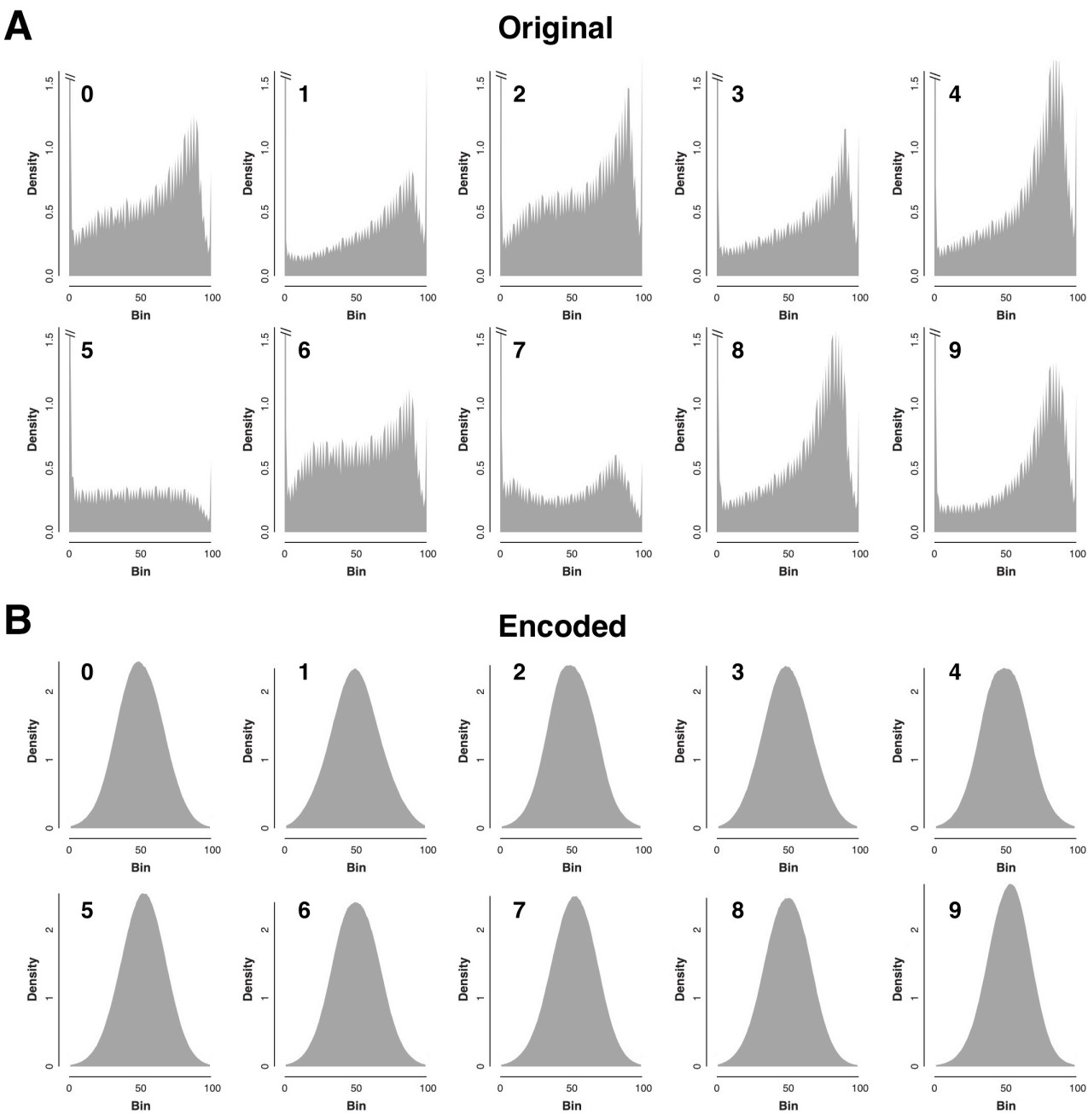

Figure 10: Average distribution of pixel values in original and encoded Fashion-MNIST training examples according to label category.

All examples are grouped according to label value in $Yd$, subjected to histogram analysis of pixel values with 100 bins, and the resulting histograms are averaged within label groups.

A. Average histograms of original Fashion-MNIST examples ($Xd$) according to label category in $Yd$, showing expected variation in pixel value distribution according to label.

B. Average histograms of transformed Fashion-MNIST examples ($Zd$) according to label category in $Yd$, showing a similar Gaussian distribution of pixel values according to label. $R^2$ goodness-of-fit $>0.997$ for the Gaussian distribution for all label categories.

variation after encoding within label category is expected from the application of random functions that are independent of labels in *Yd*.

The above analyses of pixel distribution and entropy variation both support the *a priori* assertion that knowledge of the relationship between *Zd* and *Yd* does not provide information about the random functions that are applied between *Xd* and *Zd* in a fashion that is independent of *Yd*.

Because an attacker cannot use the information leaked regarding *Yd* to gain information about the encoding random function $f(Xd) \rightarrow Zd$, an attacker's attempt to discover the inverse function $f^{-1}(Zd) \rightarrow Xd$ is probabilistically subject to the extremely large search spaces described above.

### 5.9 Encoded Dataset Challenge

An example dataset encoded with the above methods (beginning with a synthetic unknown dataset) is available at: https://drive.google.com/drive/folders/1Q0xspKzQCgP4ouYWcHGJcsLCa4o6pHjd

The encoded dataset is stored as separate .npy files for x_train, y_train, x_test, and y_test, and consists of a 60,000 train / 10,000 test split with (58 x 58) pixel grayscale examples and labels in 10 categories (0-9). There are 6,000 examples in each of the train categories, and 1,000 examples in each of the test categories. Loading the .npy filetype into Python may be performed with *numpy.load* (https://numpy.org/doc/stable/reference/generated/numpy.load.html). To match the threat model, where the attacker may be a member of Party B and thus may have access to a model trained on (*Zd*) and (*Yd*), a Keras model trained on the encoded challenge dataset (*Zd* = x_train) and (*Yd* = y_train) is also included in the same folder in HDF5 (.h5) format, which may be loaded with *tf.keras.models.load_model* (https://keras.io/api/models/model_saving_apis/).

## 6  Discussion

### 6.1  Summary

A method for secure encoding of datasets for machine learning operations is described here in detail and shows very little reduction in fully-connected ANN training performance when compared to training with unencoded datasets. The individual transformations of the method applied separately do not significantly reduce validation accuracy. Training on several datasets transformed by sequential application of padding followed by perturbation followed by orthogonal transformation results in validation accuracies that are only slightly lower than those seen with unmodified control datasets. Training time is similar between encoded and control datasets.

### 6.2  Security aspects of the method

The privacy established by our encoding methods in the context of our threat model (Figure 1) are discussed and assessed analytically in sections 5.7 'Privacy analysis' and 5.8 'Information leakage' above, with accompanying Figures 9, 10, and 11.

#### 6.2.1  Security parameters

Practical security schemes have *security parameters* that can be varied to change the strength of the encoding scheme, such as the key size in AES (National Institute of Standards and Technology, 2001). Small key sizes might allow the scheme to be easily broken with modern computing resources, while larger key sizes such as the current AES standard of 256 bit keys provide security by making the problem computationally intractable. Varying security parameters in such schemes typically involve tradeoffs; for example, larger key sizes yield greater security but make encoding computation slower.

In the present method, there are several security parameters that can be adjusted to vary the level of security for a given project within the defined threat model as shown above in Figure 1:

- Type of padding applied (*e.g.,* fixed padding, adjusted padding, non-fixed padding).

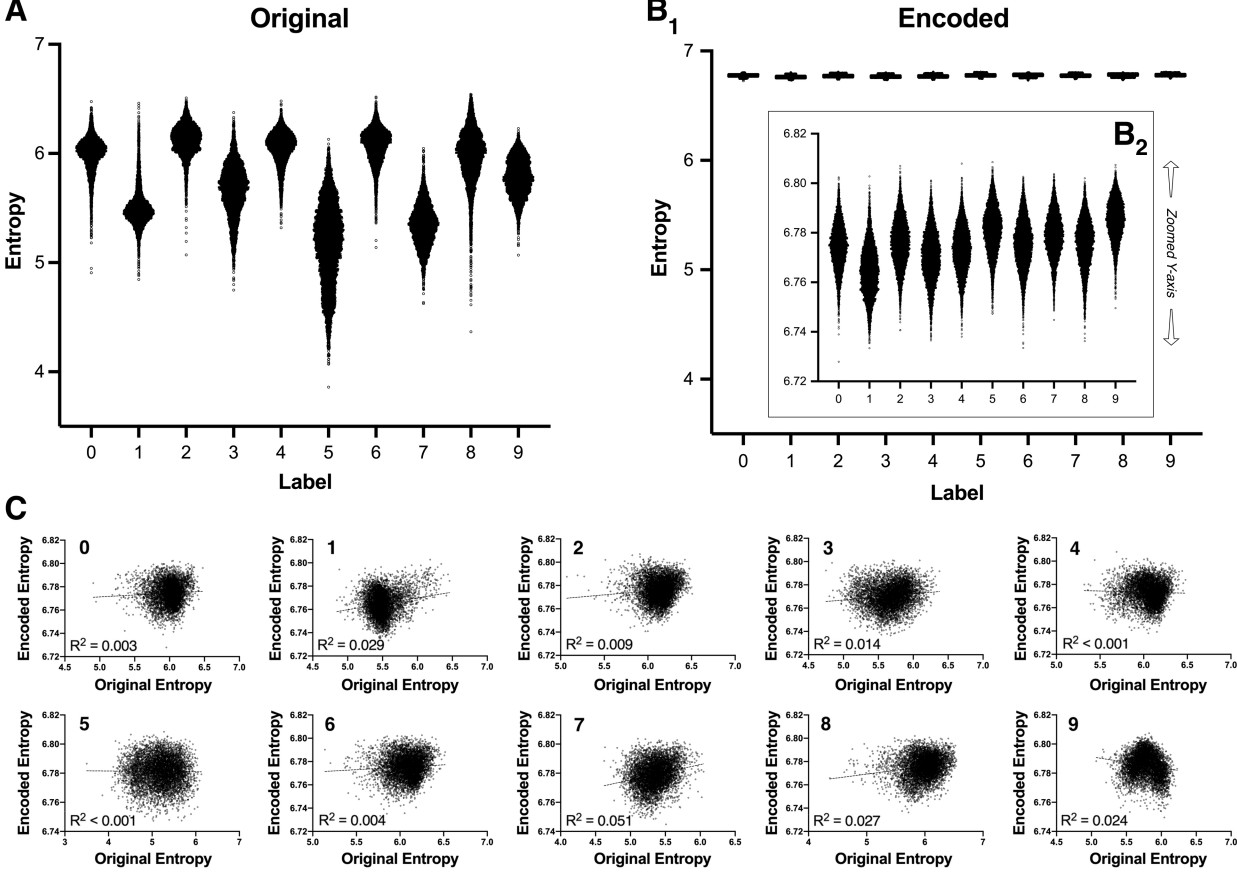

Figure 11: Entropy analysis of paired original and encoded Fashion-MNIST training examples, according to label category.

Total entropy per example calculated for each Fashion-MNIST original example in *Xd* and each encoded Fashion-MNIST example in *Zd*, grouped by label category in *Yd*.

A. Scatter plot showing the entropy distribution of original examples in *Xd* according to label category in *Yd*.

$B_1$. Scatter plot showing the entropy distribution of encoded examples in *Zd* according to label category in *Yd*, using the same Y-axis scale for entropy as shown in (A), showing that entropy increases from *Xd* to *Zd* and that *entropy variation* dramatically decreases from *Xd* to *Zd*.

$B_2$. Same scatter plot as shown in ($B_1$) but with zoomed-in Y-Axis to show variation of entropy values in each label category in the encoded set *Zd* over a much smaller range.

C. Linear regression of original entropy (*X-axis*) against encoded entropy (*Y-axis*) for paired examples before and after transformation from *Xd* to *Zd*. Best-fit regression lines in different label categories have positive, near-zero, or negative slopes, and all regressions have extremely poor $R^2$ goodness-of-fit values (*lowest* <0.001, *highest* 0.051), indicating a lack of relationship between entropy variation in the originals (*Xd*) and entropy variation after encoding (*Zd*).

- Extent of padding applied per example ($n$ pixels).
- Location of padding relative to original pixels (*e.g.,* intrinsic, extrinsic, contiguous, non-contiguous).
- Type of perturbation applied (*e.g.,* fixed perturbation, non-fixed perturbation).
- Choice of perturbation function (*e.g.,* change in pixel value by random amount or another mathematical function with a random variable).
- Perturbation range (*e.g.,* random perturbation of each pixel with bounds from -A% to +B%).
- Variation of perturbation range (where the values for A and B above are varied from pixel to pixel prior to random choice of a perturbation amount between A and B for that pixel).
- Handling of edge values with perturbation (*e.g.,* clipping, reflection).
- Choice of orthogonal transformation (*e.g.,* fixed shuffling, random orthogonal matrix transformation).
- Iterative application of multiple transformations (the ability to perform multiple rounds of padding, perturbation, and orthogonal transformation with the option to vary the above security parameters in each round).

Further detailed discussion of security in the context of our threat model are included in section 5.7 - Privacy Analysis and section 5.8 - Information Leakage, above.

Independent security testing may be performed on a publicly accessible encoded challenge dataset (see section 5.9 - Encoded Dataset Challenge, above).

### 6.3 Adaptation to convolutional neural networks

The current paper explores the performance of various aspects of the padding, perturbation, and orthogonal transformation method in the context of fully-connected ANNs. It is also possible to adapt this approach to encoding for use with convolutional neural networks (CNNs) (Anonymized, 2022). The operations of convolutions and pooling in CNNs leverage the spatial relationships of data elements, so these operations cannot be blindly performed after an orthogonal transformation such as fixed shuffling in a fashion that is agnostic to the fixed shuffling algorithm. Instead, convolution and pooling operations may be performed prior to application of padding, perturbation, and orthogonal transformation, with tensors created that store multiple encoded results of convolution and pooling (Anonymized, 2022). Alternatively, the algorithm used to apply padding, perturbation, and an orthogonal transformation such as fixed shuffling may be encrypted by conventional methods, transmitted securely, then decrypted by authorized users to apply convolution and pooling operations, using the decrypted algorithm as a lookup table that enables these operations on encoded data (Anonymized, 2022). Such integration of our encoding method with CNN structures is beyond the scope of this initial report.

### 6.4 Alternative approaches

Various alternative approaches to the problem of securing data for model training have been proposed, with recent work focused on the areas of *federated learning*, the use of data treated with *fully homomorphic encryption*, and several proposals for instance encoding, including the InstaHide method (Huang et al., 2021), the NeuraCrypt method (Yala et al., 2021), the DarKnight scheme (Hashemi et al., 2020), and application of single-step orthogonal transformation based on an image key (Chen, 2019).

#### 6.4.1 Federated learning

Federated learning is a well-established approach for performing distributed model updates using data on local devices, wherein training data do not have to be transmitted from the local device to a central server. In the federated learning framework, a model and weights are transmitted to the local device, weights are updated by additional local training, and then the updated models and weights returned to the central server (McMahan et al., 2017; Kairouz et al., 2021; Li et al., 2020; Rodríguez-Barroso et al., 2020). While this approach has the advantage of not needing to transmit data from a local device to a central server, and has found wide application in updating models using rich data on more computing-constrained systems such

as mobile devices, it has certain disadvantages when it comes to the problem of model training of private data. Software provided by the central server must be run to train or update models on local systems that maintain the private data, and this software has direct access to the private data. This means that the party maintaining the private data ('Party A') must trust the party providing the software ('Party B'). Party A must trust Party B to run software behind the Party A's firewall and additionally trust this software to have direct access to Party A's private data. Party A additionally must trust Party B not to transmit any aspect of Party A's private data back to Party B's servers along with the updated models and weights that have to be transmitted from behind Party A's firewall to Party B's server.

In addition to the security issues discussed above, federated learning can have an important adverse business ramification: because federated learning is particularly well-suited for updating models on a wide range of data sources, this means that the business value of private data provided by any one organization will tend to be devalued by dilution.

### 6.4.2 Fully homomorphic encryption

Fully homomorphic encryption (FHE) represents a family of approaches that allow mathematical manipulation of encrypted data such that when a manipulated ciphertext is decrypted, the result is identical to an analogous mathematical manipulation performed on the original data prior to encryption (Lauter, 2021; Munjal & Bhatia, 2022; Alloghani et al., 2019). There have been initial explorations of using FHE in the context of secure machine learning (Bost et al., 2014; Graepel et al., 2013; Lauter, 2021; Nandakumar et al., 2019; Papernot et al., 2016), but FHE requires profound alterations to machine learning algorithms that yield compute times that are many orders of magnitude higher than required for work on unencrypted data using conventional methods (Nandakumar et al., 2019). For example, training on MNIST in an initial exploration of using FHE with a neural network required downsampling of the MNIST dataset to 8x8 pixel examples and use of a cyclotomic ring encryption considered to not provide sufficient security ($\phi(m) = 600$), and despite these simplifications, compute times for the neural network ranged from 40 minutes to 1.5 days (Nandakumar et al., 2019). When a more secure FHE algorithm was tested ($\phi(m) = 27,000$), the time to process a single neuron in the first layer was 15 minutes (Nandakumar et al., 2019). In another experiment to perform unsupervised learning with K-means-clustering on an FHE-encrypted dataset, single-thread runtimes were estimated to be on the timescale of months (Jäschke & Armknecht, 2019).

### 6.4.3 The InstaHide method

One form of instance-hiding encoding designed for private machine learning that has been proposed recently is the InstaHide method (Huang et al., 2021). InstaHide first applies the approach of *mixup* (Zhang et al., 2018), performing a linear combination of several images from the train set and a large public database to generate multiple examples derived from combinations of the original examples. Next, the InstaHide method extends image pixel values to a negative-to-positive range, then applies a random pattern of sign inversions to each image (Huang et al., 2021). Subsequent work by others (Carlini et al., 2021a) has shown this method to be vulnerable to an attack that takes advantage of the specific steps known to occur in the InstaHide method, in particular, taking the absolute value of all pixel values (to remove the sign-flipping in InstaHide), cluster encoding the dataset by training a neural network "to detect when two encodings were generated from the same image," and solving an under-determined system of equations by gradient descent to recover an approximation of the original images (Carlini et al., 2021a). The specific sources of randomness in InstaHide that are the basis of this method's claim to privacy allowed for the attack by Carlini, et al.: performing linear combinations using multiple images, in turn yielding multiple images, and performing a simple random alteration (random sign flipping) (Carlini, 2020).

The method applied by InstaHide, and the corresponding attack by Carlini, et al., are in several important ways unrelated to the approach we take in this paper. First, InstaHide uses linear combinations of multiple images, including images from a large public dataset, to yield encoded images. This level of complexity actually creates the weakness exploited by Carlini, et al., as they point out in a related blog post (Carlini, 2020). It is this linear combination of images that allows the second step (cluster encoding) to "detect when two encodings were generated from the same image" (Carlini et al., 2021a). Our approach does not have this attack vulnerability because it does not in any way combine multiple images, which is necessary to enable

this step of the Carlini attack. More trivially, the specific transformation of applying random sign inversions as used in InstaHide and nullified by Carlini by application of the absolute value of all pixels in the first step of their attack is also not relevant to our approach. In our approach, every example in the original dataset has a single corresponding example in the encoded dataset, and there are several sources of randomness used to alter the data, with all sources of randomness independent of label category and kept secret from a potential attacker.

### 6.4.4 The NeuraCrypt method

Another form of example encoding for private machine learning proposed recently is the NeuraCrypt method (Yala et al., 2021). NeuraCrypt performs random encoding of an original dataset by sampling a random neural network with convolutional layers, batch normalization, and ReLU, using this randomly chosen network to encode the data, together with random positional embedding and permutation of image patches, to "encode positional information into the feature space while hiding spatial structure" (Yala et al., 2021). The architecture of the networks used in NeuraCrypt is "closely inspired by the design of patch-embedding modules" of Vision Transformer networks (Dosovitskiy et al., 2021).

In a subsequent paper showing that it is possible to fully break the privacy of the NeuraCrypt method, the attack leverages the particular structure of NeuraCrypt, namely the use of patch-embedding and information leakage from the position encoding used (Carlini et al., 2021b).

### 6.4.5 The DarKnight method

Another proposed approach to training sensitive data using untrusted cloud hardware is the DarKnight approach, which passes traditionally encrypted examples to the cloud, where the examples are decrypted within a trusted execution environment (TEE) and basic ML operations are begun within this TEE (Hashemi et al., 2020). In order to perform intensive linear and nonlinear operations on untrusted cloud GPUs, the DarKnight scheme blinds the data passed outside the TEE to the untrusted hardware as a *virtual batch*, where a number of input examples are linearly combined to form a number of encoded examples (Hashemi et al., 2020), similar to the linear combination *mixup* approach used as part of the InstaHide method discussed above (Huang et al., 2021). It is possible that one aspect of the attack used by Carlini, et al., on InstaHide, leveraging cluster encoding to detect linear combinations including the same example (Carlini et al., 2021a), might represent a potential attack vector against DarKnight as well, but to our knowledge this has not been explored to date.

### 6.4.6 Single-step orthogonal transformation

Orthogonal transformation has been proposed as a one-step encoding for machine learning applications, specifically by performing QR factorization on a digital image file of appropriate size and using the resulting orthogonal matrix to transform input vectors into encoded vectors (Chen, 2019). Using an orthogonal matrix to transform input vectors into encoded vectors maintains the angles between vectors and the lengths of vectors, by transforming the relative position of the vectors in the coordinate system (representing a rotation and/or reflection of the coordinate system, or in the specific case of index shuffling, shuffling of the coordinate axes). Because the relationships between vector angles and vector lengths are preserved in this transformation, machine learning accuracy is expected to be unaffected, and we show here that this is indeed the case in practice.

Specific approaches are available to obtain an orthogonal matrix for use in encoding, and these approaches have important differences. Random sampling of orthogonal matrices from the O(N) Haar distribution yields a random orthogonal matrix, while random fixed shuffling yields a random selection of an index shuffle within the much broader space of orthogonal matrices. The process of QR factorization of a starting non-orthogonal matrix to yield an orthogonal matrix and an upper triangular matrix yields a random orthogonal matrix *if the starting matrix is chosen randomly*, which is not the case when starting with an image file. If one starts with a random matrix and applies QR factorization, the random orthogonal matrix resulting from this process is random, but is drawn from a distinct distribution within the overall space of potential orthogonal matrices. Finally, the choice of method for generating the orthogonal matrix for encoding has practical

implications, as sampling the Haar distribution and QR factorization of a random matrix are roughly cubic in time relative to input vector dimensions, while index shuffling is roughly linear in time and may be more appropriate for larger examples.

In our present method, we use the more general approaches to random orthogonal matrix generation (e.g., sampling from the Haar distribution or index shuffling) and further strengthen privacy by combining one or more of these orthogonal transformations with the non-orthogonal and nonlinear transformations of padding and perturbation, yielding an aggregate non-orthogonal and nonlinear transformation that changes vector dimensions between *Xd* and *Zd.*

### 6.4.7   Other approaches

Other variations on the theme of homomorphic encryption with additional constraints have been tested with neural networks with improved performance results, such as the approach of *leveled homomorphic encryption* (Bos et al., 2013), together with tailoring of network structures to the specific encryption approaches used in order to make the required computations more tractable for neural network training (Gilad-Bachrach et al., 2016).

Another proposed approach adapts multi-party computation techniques (Goldreich, 2004), in which encrypted data is passed back and forth between parties, updating calculations layer by layer in stepwise fashion, with the sending party required at each step to decrypt the data, apply a nonlinear transformation, encrypt the result and return this to the other party (Barni et al., 2006; Orlandi et al., 2007). While recent adaptations of this technique have improved the computational efficiency, the approach still requires bidirectional transmission of very large amounts of data, substantially longer training times when compared with plaintext training, and technical sophistication on both ends of the data exchange to implement the method (Keller & Sun, 2021).

Finally, a broad area of computer science research deals with the topic of *privacy-preserving data analysis* (the attempt to achieve a balance between data utility for analysis, particularly in the aggregate, while maintaining privacy, particularly at the individual level. The notion of *differential privacy* is a modern approach to privacy-preserving data analysis that quantifies the tradeoff between data utility and privacy when training on original, sensitive data (Dwork & Roth, 2014). The goal privacy guarantee in differential privacy is that an attacker does not learn significantly more about an individual (or example) if that individual (or example) is present in a training dataset than when they are *not* present in the training dataset (Dwork et al., 2006).

### 6.5   Advantages of the presented methods

Compared to alternative approaches, the method presented here has certain advantages. The random encoding process described here, when used with fully-connected ANNs, allows for use of unchanged machine learning systems, without need for fundamental alteration of existing frameworks. The method also allows for customization of the encoding process, with the option to vary multiple different security parameters, so that the trade-off between security and small decrements in accuracy may be adjusted according to a given project's needs. There is no loss in training time when using the encoding described here, and only minimal one-time computations are needed to perform the encoding steps.

Importantly, the specific encoding steps used are applied in similar fashion to the training and test tensors and any inference examples. This requirement to encode inference data protects the business value of the encoded training data used in a collaboration, because the collaboration may be set up such that the party that contributes encoded data maintains the secret of the encoding process, which must be securely accessed in order to perform inference.

### 6.6   Conclusion

A technique for securely encoding examples with non-orthogonal and nonlinear transformation is presented that enables direct ANN training on encoded datasets. When used with a non-convolutional, fully-connected ANN as shown in the present report, no modifications to standard neural network architecture are needed.

Individual, single-step transformations of the method do not significantly reduce validation accuracy, and training on datasets transformed by sequential padding, perturbation, and random orthogonal transformation yields only slightly reduced validation accuracies. No differences in training times are seen between encoded and control datasets. Because a potential attacker cannot use the information leaked in our threat model regarding the relationship between $Zd$ and $Yd$ to gain information about the encoding random function $f(Xd) \rightarrow Zd$, an attacker's attempt to discover the inverse function $f^{-1}(Zd) \rightarrow Xd$ is probabilistically subject to extremely large search spaces. The methods described here may useful in a broad range of machine learning projects requiring data security.

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
