# OpenReview forum: "Direct Neural Network Training on Securely Encoded Datasets"
_TMLR — Rejected by TMLR_

### Review · Reviewer_pTas · 2022-10-31

**Summary Of Contributions:**

Contributions the paper claims:

(1) This paper presents a technique that enables the training of neural networks without revealing the original data instances. (2) The paper employs several input transformations, such as padding, perturbation, and/or orthogonal transformation, to encode the input instances. (3) The paper shows that, while those encoding schemes hide the details in the input space, they do not reduce the accuracy of neural networks trained on the data.

**Audience:**

Yes

**Broader Impact Concerns:**

This work can harm the public by proposing a privacy scheme that was studied and already broken by the community.

**Claims And Evidence:**

No

**Requested Changes:**

No adjustment request; the problems that I mentioned above cannot be addressed by revisions.

**Strengths And Weaknesses:**

Strengths

1. The paper empirically studies the impact of input encoding schemes on the performance of neural networks trained on the encoded data


Weaknesses

1. The paper proposes a scheme that was already broken by the prior work
2. The work potentially misleads readers and makes them vulnerable to privacy attacks
3. A series of input transformations are not technically novel
4. The paper does not evaluate the privacy aspects of the proposed scheme at all


Detailed comments:

This paper showed the idea that, by transforming the input data, we could achieve high accuracy on the data while preventing some privacy risks. Unfortunately, those directions have been studied extensively by prior work. Huang et al. [1] studied the same direction--claiming that we could preserve the privacy of data by instance encoding, which uses much more advanced input transformations. Carlini et al. [2], in contrast, showed that such a scheme can be broken "easily" and provide zero privacy guarantee.

As this paper:
(1) is "largely" detached from the prior work and
(2) can make readers use the "broken" privacy preserving scheme,
I highly recommend to reject this paper.

[1] InstaHide: Instance-hiding Schemes for Private Distributed Learning
[2] Is Private Learning Possible with Instance Encoding?

To add a bit more:

On the technical side, the impact of input transformations has been studied under the area of "data augmentations," and it's not at all surprising that we can train a neural network (it's not even a convolutional network) without causing a significant accuracy drop in MNIST.

Perhaps, based on my results from the prior work, the fact that the paper doesn't consider larger datasets like CIFAR10 or ImageNet makes it leads to this conclusion. If we practice the same strategy with larger datasets, I am sure we will observe a significant accuracy drop. For example, we could see that adversarial training drops the accuracy of a CIFAR10 model by 10-15%. I don't think the actual drop by the transformation would be higher like that, but I am sure it will be more than 5% (as we observe a 1-2% drop in the MNIST models).

Moreover, this paper did not propose any new input transformations, but just employed and combined existing techniques.

Last but the "most" important, this paper claims that the proposed scheme will improve privacy, which completely invalidates the main contribution. However, no experiment is there to evaluate this claim. I would recommend using the attacks proposed by [2] to claim privacy.

---

> ### Author Response · Authors · 2022-12-22
> **Response to Reviewer pTas**
>
> We appreciate all of the reviewers’ obvious expertise in this subject area and their thoughtful responses to our manuscript.
>
> Reviewer pTas's main criticism:
> - InstaHide is a type of instance encoding;
> - Carlini, et al., broke InstaHide;
> - All forms of instance encoding are thus vulnerable;
>
> InstaHide uses linear combination of multiple images plus random sign inversions. Carlini’s attack on InstaHide is:
> - Take the absolute pixel values;
> - Cluster encode the dataset with a neural network “to detect when two encodings were generated from the same image.”
> - Recover images by gradient descent.
>
> InstaHide and the Carlini attack are unrelated to the approach we take. InstaHide uses linear combinations of multiple images, which we do not do, and we do not apply random sign inversions.
>
> Below please find point-by-point responses, with our responses [  flanked by brackets  ].
>
> Weaknesses: The paper proposes...
> [  Our approach is completely distinct from the approach used by Huang, et al., and thus the attack provided by Carlini does not apply to our approach. It not correct to state that our approach has been broken by prior work. Our revised paper now includes a detailed description and reference to InstaHide, NeuraCrypt, and the Carlini, et al., papers breaking each of these approaches, as well as a full discussion of how these approaches are distinct from ours.  ]
>
> The work potentially...
> [  We have expanded our description of the threat model, information leakage, and added analysis showing that the information leaked regarding the association between encoded examples and labels does not provide information that is useful to reconstruct or reverse the random, label-agnostic, encoding functions we apply, in new Section 2 ‘Threat model’, 5.6 ‘Testing on alternative data sources’, 5.7 ‘Privacy analysis’, and new Figures 1, 9, 10, & 11, as well as a reworked discussion subsection 6.2 ‘Security aspects of the method’. ]
>
> A series of input transformations...
> [  We would hope that the novelty of our specific methodology, as described in our paper in more detail after revision, be considered on its own merits. ]
>
> The paper does not evaluate...
> [  Our discussion of the privacy aspects of our method were in need of substantial improvement. We have now added a concrete threat model and privacy definition with analysis of information leakage, in the above mentioned additions.  ]
>
> Detailed comments: This paper showed...
> [  See above re: InstaHide and the Carlini attack. ]
>
> As this paper...
> [ See above re: InstaHide and the Carlini attack. We greatly appreciate the feedback provided by this reviewer, which have strengthened the connection of our manuscript to the prior literature.  ]
>
> On the technical side...
> [  Where the reviewer says “we observed a 1-2% drop in the MNIST models,” this is unclear: we see a 1% decrement in accuracy (see, for example, Figure 6C and the new Table 1). With regard to the prediction that we might see a drop of 10-15% with CIFAR-10, we have now performed analyses on CIFAR-10 and find a 2.16% drop in accuracy. Additional training on Fashion-MNIST and on our synthetic unknown challenge dataset are also added to the new section 5.6 - ‘Testing on alternative data sources’ and a new Table 1 ]
>
> Moreover...
> [  The reviewer suggests that our methods are not novel, because they are a series of input transformations. There is a parallel here to the issues discussed above—the details of our approach are not considered on their own merits because they are viewed in terms of a broad category of approaches, a subset of which have been explored in the literature.  ]
>
> Last but the...
> [  Based on the feedback from this reviewer and the other reviewers, we have added a better definition of the threat model and privacy claims, with new including addition of Section 2 ‘Threat model’, 5.7 ‘Privacy analysis’, 5.8 ‘Information leakage’, and new Figures 1, 9, 10, & 11, with a reworked discussion subsection 6.2 ‘Security aspects of the method’, which emphasizes the security parameters in the method in a new subsubsection 6.2.1 ‘Security parameters’. It should be noted that we cannot apply the specific attacks proposed in the Carlini, et al., reference ([2]), for the reasons explained above - these attack approaches do not apply to our technique as they specifically target core aspects of the Huang, et al., approach that are not present here. In the context of a clear definition of the threat model and the associated information leakage, we now show, in 5.7 ‘Privacy analysis’ and Figures 9, 10, & 11, that the information leaked (the association between encoded examples and labels) does not simplify the problem of attempting to reverse the label-agnostic random functions that are applied between original and encoded images. ]

---

> > ### Comment · Reviewer_pTas · 2023-01-17
> > **Thanks for the Response**
> >
> > I thank the authors for the detailed response.
> >
> > I want to clarify the point that I am making in my review. It's **NOT** that "Carlini et al. breaks InstaHide; thus, all the encoding schemes can be broken." The response misunderstood the paper's point by Carlini et al.
> >
> > Suppose that we want to run neural network inference on an encoded dataset. Two approaches possible so far:
> >
> > 1. Homomorphic encryption.
> > 2. Computing on "encoded" datasets.
> >
> > Homomorphic encryption "encrypts" the data and runs computations on it. It's secure against the adversary this paper assumes, but it's out of this manuscript's scope. So, let's exclude this.
> >
> > So let us take a look at the second approach.
> >
> > The objective of the second one is:
> > 1.  we do not want to encrypt the data, but
> > 2. we want to hide some details in the data so that an adversary cannot "visually" recognize what it is, and
> > 3. we want to train a network on it without hurting the accuracy of the trained model compared to the vanilla one.
> >
> > Carlini's paper claims that **objectives 1 and 3 are contradictory.**
> >
> > > If we train a model without any accuracy drop on "an encoded dataset, "then it means the data (anyway) contains some statistical properties useful for satisfying the target task. With a sufficient amount of computations and knowledge, the adversary can always reconstruct the original data. That is the point.
> >
> > One question we might ask: "but we make the reconstruction XYZ times harder than before."
> >
> > So the same paper claims that **such confidentiality should be evaluated carefully**.
> >
> > > (1) In cryptography, when we evaluate the strengths of an encryption scheme, we evaluate how difficult it would be to "reconstruct" the original texts. It is the same for the proposed method unless the paper is not about privacy at all. In that sense, unless the paper evaluates against some reconstruction attacks, we do not know whether the proposed method improves confidentiality.
> >
> > > (2) So, when evaluating such confidentiality, the work also recommends evaluating against the strong adversary. Normally, in cryptography, we assume that the adversary will have a strong knowledge of the victim's systems, such as the adversary with the partial knowledge about the training data, encryption schemes, etc. But a vast literature has shown that only the randomness matters for the strong defense. As we discussed "the scheme allows a network to pick some properties from the encoded data to have a sufficient accuracy", the attacker can also do the same or at least similar with high precision.
> >
> > I appreciate the authors putting a lot of work into writing and revising this paper, but the underlying assumption and the way the method is evaluated seem incorrect. Thus, I would keep my evaluation score.

---

### Review · Reviewer_XEeZ · 2022-11-23

**Summary Of Contributions:**

This paper studies the problem of data encoding for privacy-preserving image classification. After motivating the problem at length in the context of data analysis for medical applications, this work presents a method for encoding sensitive data that is claimed to preserve the utility for data analysis. The rationale behind this scheme is to use successive (possibly random) transformations that individually do not affect much the validation accuracy of the model. While focusing on three types of transformations, namely padding, perturbations, and orthogonal transformation, the paper provides a thorough analysis of the impact these transformations have on the validation accuracy of a simple feedforward neural network for MNIST image classification task. To the best of my understanding, no security nor privacy guarantees are provided for the scheme. The paper provides a link to an encoded dataset  and encourages the community to test the reliability of the proposed encoding scheme in terms of image protection.


**Audience:**

No

**Broader Impact Concerns:**

I have no particular broader impact concern, besides the above requested changes.

**Claims And Evidence:**

No

**Requested Changes:**

### Suggestions that would simply strengthen the work

1. Better highlighting of contributions
2. Consider differential privacy as related work

### Suggestions critical to securing recommendation for acceptance

1. Augment existing results by considering more difficult benchmark tasks in image classification (e.g., CIFAR or ImageNet) or by considering medical datasets.
2. Introduce a clear threat model and provide a theoretical or empirical analysis on the reliability of the proposed encoding scheme.
3. Incorporate a discussion of the similarities and differences with existing methods similar to Instahide and Neuracypt.

**Strengths And Weaknesses:**

### Strengths

The problem of designing encoding schemes for private data analysis is an old but highly relevant topic in computer science and healthcare. As machine learning becomes ubiquitous in many aspects of medical data analysis, it is critical to continue to design efficient methods that protect patient privacy without compromising the predictive quality of the system. The motivation section of the article really brings this point forward and highlights very well the critical nature of the problem considered. This goes along with the fact that, in my opinion, the paper is overall quite well written. I would also like to point out that the paper gives a lot of details and explanations on the different methods that are being tested as well as the experimental setup that is being considered.


### Weaknesses

In my opinion, this work has three main weaknesses (one minor and two major), as described below.
- **Contributions are not clearly highlighted (minor).** While the paper is overall well written, I think the contributions are not highlighted in a very natural way. I explain this in two points below.
  - As I understand it, the contribution of the paper is the introduction and analysis of a new encoding scheme that, when coupled with a feedforward network, achieves performance similar to what can be obtained on an unencoded dataset. However, the first time this claim is clearly made is in Section 5.1 (page 13).  I believe that the clarity of the paper would greatly benefit from a reworking of the last paragraph of Section 1 to present not only the proposed/studied methods but also the main conclusions of the study, i.e., the main technical contributions.
  - I was also a bit puzzled by the fact that the article cites an anonymous reference *'(Anonymized, 2022)'* several times, especially when presenting the solutions. I assume that this reference is anonymized because it comes from the same authors as the current article, however TMLR does not accept extensions of already published work. Therefore, it seems essential to me that the authors comment on the differences of the current submission with this previous work. Note that it is unclear to me whether this rule applies to patents, if it does not, please ignore this comment.

- **MNIST is not a good benchmark for empirical studies (major).** I was quite concerned that the only task the article addresses is MNIST, mainly for two reasons. First, the entire story of the paper revolves around the analysis of medical data. Thus, it would have been more consistent to consider a dataset containing medical images or data (see for example https://www.kaggle.com/general/168211). Second, and more importantly, I do not think that the MNIST task should be used as the sole basis for claiming a scientific contribution. In fact, MNIST is a very simple task that has been shown to be easily solvable, even in the presence of noise and/or perturbations. Hence, I believe that MNIST can be used as a support dataset to complement theoretical results and/or other experiments on more difficult tasks, but it should not be used as a stand-alone task to prove a point. I recommend augmenting existing results by considering more difficult benchmark tasks in image classification (e.g. CIFAR or ImageNet) or by considering medical datasets.

- **No security/privacy analysis (major).** As I understand it, the paper suggests that the proposed scheme is safe because it reduces to a *'nonothogonal and nonlinear aggregate transformation'*. This argument is emphasized in Section 5.2 where we get a high-level intuition on why the method protects the images from reconstruction. However, the article does not present any concrete security analysis, either theoretical or empirical. In fact, the paper does not even present the threat model that the encoding method is supposed to address. In my opinion, the fact that no security aspects are clearly discussed makes it impossible to assess the reliability of the method from a security and privacy perspective. This seems even more critical to me as the method appears to be close to a recent line of research which attempts to encode image datasets using neural networks or arbitrary transformations and which provides questionable guarantees (see e.g., https://arxiv.org/abs/2011.05315 and https://arxiv.org/abs/2108.07256). I recommend introducing a clear threat model in the paper and providing a thorough analysis (theoretical or empirical) on the reliability of the proposed encoding scheme. I also suggest incorporating a discussion on the similarities and differences that the proposed method shares with the abovementioned line of research (particularly with respect to *Instahide* and *Neuracypt*). Finally, I strongly encourage the author(s) to consider differential privacy (https://www.cis.upenn.edu/~aaroth/Papers/privacybook.pdf) as a related line of research to build upon or to simply mention in the related work section. Differential privacy has become one of the gold standards in privacy-preserving machine learning. It provides both theoretically sound security/privacy results and reasonably good accuracy. I will not elaborate on this, but I think it is at least worth mentioning when considering encoding schemes for private data analysis.

---

> ### Author Response · Authors · 2022-12-22
> **Response to Reviewer XEeZ**
>
> We appreciate all of the reviewers’ obvious expertise in this subject area and their thoughtful responses to our manuscript. Here we respond point-by-point to the reviewer’s comments, with the reviewer comments in plain text and our responses [  flanked by brackets  ].
>
> Weaknesses: ...
> [  We have now introduced a clear statement to the last paragraph of the Introduction section as suggested. ]
>
> I was also...
> [  The reviewer is correct: this reference is anonymized because listing the authors of the reference would identify the authors of the present work and break anonymity. The reference is a patent application and it does not show any of the analyses presented here comparing network training between control and encoded datasets—all of the work in the present manuscript is new and does not duplicate the work referred to in the anonymized patent application.  ]
>
> MNIST is not ...
> [  We understand the reviewer’s perspective on MNIST vs harder ML datasets, as well as their suggestion that our methods should be tested on more than one dataset. To test our encoding process with other datasets, we have now added new analyses to the paper using Fashion-MNIST, CIFAR-10, and our synthetic unknown challenge dataset, under a new subsection 5.6 ‘Testing on alternative data sources’. Using CIFAR-10, as requested by all three reviewers, a similar fully-connected neural network to those used throughout our paper shows a drop of about 2% in absolute validation accuracy between control CIFAR-10 and CIFAR-10 encoded by non-fixed padding, non-fixed perturbation, and fixed shuffling. This relatively small decrement is observed despite the fact that using CIFAR-10 to train a non-convolutional network is well-known to yield a low level of baseline accuracy (control 0.5505 + 0.0174 vs encoded 0.5289 + 0.033; P < 0.001). With Fashion-MNIST, which achieves intermediate levels of control accuracy with a fully-connected ANN between those of MNIST and CIFAR-10, we find that validation accuracy is 0.8999 + 0.0032 for control, and 0.8874 + 0.0025 when encoded by non-fixed padding, non-fixed perturbation, and fixed shuffling, P < 0.001. Lastly, for our synthetic challenge dataset, we find a decrease in accuracy of just over 1% between control (0.9849 + 0.0016) and encoded (0.9726 + 0.0028), P<0.001.
>
> Also of note, while it is possible to integrate our method with convolution and pooling operations, to do so would be beyond the scope of this initial report. As described in the other Response to Reviewer comments, we have further clarified in multiple places in the manuscript that our results in the present paper only pertain to experiments with fully-connected ANNs.  ]
>
> No security/privacy analysis...
> [ It was clear that we needed to substantially improve our description of the privacy definition and threat model, as requested by this reviewer and the other reviewers. We have now added a new section 2. ‘Threat model’ with accompanying new Figure 1 to more clearly delineate and discuss the threat model and privacy implications. Our analysis sections and discussion (including 5.7 ‘Privacy analysis’, 5.8 ‘Information leakage’, and new Figures 9, 10, & 11) now better address the privacy implications of our encoding process in the specific context of this threat model, showing both theoretically and by analysis that the information leaked in our threat model (the association between encoded data and labels) does not provide information that would be useful to attempt to reverse the random, label-agnostic, functions applied between the original and encoded examples. Finally, a reworked discussion subsection 6.2 ‘Security aspects of the method’ now better emphasizes the security parameters in our method. ]
>
> I also suggest...
> [ As this reviewer and the other reviewers refer to two schemes that have recently been proposed (InstaHide and NeuraCrypt) and then later broken by Carlini, et al., we have added detailed descriptions of InstaHide and NeuraCrypt and detailed how our privacy-protecting encoding differs from these approaches and is not vulnerable to the published attacks presented by Carlini, et al. ]
>
> Finally, I strongly...
> [ We appreciate this suggestion, and we have now added a brief discussion of differential privacy, including the above reference, to what we feel is now a better-rounded discussion based on reviewer feedback. ]
>
> Requested Changes: Better highlighting...
> [ See our responses and manuscript additions / changes as detailed above. ]
>
> Suggestions critical...
> [ See our responses and manuscript additions / changes as described above, including addition of Section 2 ‘Threat model’, 5.6 ‘Testing on alternative data sources’, 5.7 ‘Privacy analysis’, and new Figures 1, 9, 10, & 11, a reworked discussion subsection 6.2 ‘Security aspects of the method’, which emphasizes the security parameters in the method, as well as a new Table 1. ]

---

> > ### Comment · Reviewer_XEeZ · 2023-01-15
> > **Final comment after updates**
> >
> > I would like to acknowledge the efforts made to improve the paper. Many of my comments have been taken into consideration, however, I still do not feel comfortable recommending acceptance at this point due to the lack of systematic analysis of the security aspects of the method.
> >
> > In my initial review, the second point that was reportedly critical in obtaining my acceptance recommendation was the introduction of a clear threat model and a theoretical or an empirical analysis on the reliability of the proposed encoding scheme. Although efforts have been made to improve the presentation of the threat model, in my opinion, the goals the method is trying to achieve in terms of security are still not clear enough. In short, for a security claim to be made, the paper must present a clear definition of what is considered a security breach/information leak (as also pointed out by reviewer F9qK). As a result, I also do not consider the sections that make security claims to be compelling enough to accept the paper.
> >
> > A quick note: The reason I suggested comparing this work with differential privacy in the first place is exactly because in that community the definition of the security objective is clear and formally defined. I suggest taking inspiration from these kinds of definitions to introduce the defender goals in future versions of the paper.

---

### Review · Reviewer_F9qK · 2022-11-24

**Summary Of Contributions:**

The paper proposes an approach that allows for neural network (NN) training and inference on encrypted data. The main idea is to introduce a sequence of transformations, that serve to mask the data, is hard to invert, and still enables training and inference with very small loss in model utility and run time. While the general goal is laudable, I think the paper has several major issues, as discussed in the rest of this review.

**Audience:**

Yes

**Broader Impact Concerns:**

I think advocating purely heuristic security and privacy claims, without any clear notion of what these mean, is generally a very harmful direction, and has the potential to undermine the entire research field of secure and privacy-preserving machine learning.


**Claims And Evidence:**

No

**Requested Changes:**

Address the issues mentioned in the weaknesses. Most importantly, clarify what is meant by security, and what kind of security guarantees does the proposed approach have (issues 1-3). Additionally (and less importantly), show that the proposed approach works with harder datasets and different models (issue 4).

**Strengths And Weaknesses:**

### Strengths:
1)  The general goal of enabling arbitrary NN training with encrypted data is important.
2)  The paper is mostly nicely written and easy to understand.


### Weaknesses:

There are several major shortcomings in the paper, and many of the claims are either over-ambitious or too vague. The most important claims are not well-supported by theory or by experiments. The following is a more detailed list of the problems:

1) The most critical concept in the paper is secure encoding, but this is not clearly defined anywhere. Therefore, none of the claims about the proposed approach providing privacy and/or security can really be shown to be true or false. Typically, secure computation requires, e.g., that a well-defined adversary (for example, honest but curious or malicious, which is computationally bounded) provably learns nothing but the final result of the computation with high probability, that depends on some security parameter. In contrast, the security of the proposed approach rests entirely on heuristic arguments; it is not clear if the proposed approach should guarantee complete obfuscation for every pixel in an image, if the aim is to protect essentially the label of an image, or something in between.

2) Related to the fuzziness of the security definition treated in 1),
the hardness of attacking the proposed transformations is very unclear: since there is no clear definition of what it means to break the proposed approach, there is also no clear way to show that it is secure or insecure.
Instead, heuristic arguments, such as the huge size of the space of all possible transformations (see Section 5.2), is used to argue that inverting the transformation is infeasible by brute force.
But as is also clear from the discussion in Section 5.2, depending on the input and on the specific transformation used, exactly inverting the transformation might also be almost trivial.
This issue is made a lot worse, if the the attacker happens to have information about any specific input (compare to standard semantic security).
There is no security parameter that defines how hard an attack will be, since everything depends on the specifics of the input and of the exact transformation used.
Issues 1) and 2) mean that the security claims in the paper are very much not supported (compare e.g. to claims in Section 5.6).

3) For most of the paper (see e.g. abstract, introduction, Section 2), the proposed approach is marketed as enabling training and inference on general NNs, while the full proposed approach actually only works with methods such as fully connected NNs, that do not use any information about the problem topology (e.g. shuffling pixel values obviously destroys any information about the pixel neighbourhoods). While simple fully connected networks work with small problems like MNIST, this is a severe limitation that should be made clear from the beginning.
While the authors propose separate treatment for convolutions, this also makes it clear that the method will not work on general NNs without important extra assumptions and modifications (compare this e.g. to the claims in Sections 5.5 and 5.6).

4) Related to 3), the experimental evidence for the proposed approach is very limited: testing on MNIST alone does not provide much evidence for the proposed approach, since it is too simple and small dataset. To show that the approach really does work, consider using harder datasets, such as ImageNet or at least CIFAR, and more varied models.


5) The alternative approaches section 5.4 misrepresent some parts and are missing citations:

5.1) There are several existing research directions and papers that have similar aims as the current paper.
For example, InstaHide of Huang et al. 2020 uses similar heuristic arguments  as the current approach (also, much of the criticism directed at InstaHide applies to the current paper, see Carlini 2020),
while a completely missing research direction trains NNs securely using secure enclaves (see e.g. Hashemi et al. 2020).

5.2) Some of the claims on federated learning (FL) seem strange. For example, it is mentioned as a problem that software for FL needs to be trusted. How is this different from what would happen with the proposed approach, or do you suggest that each party using the proposed approach independently implements it? This is a problem generally solved, e.g., with open source code, digital signatures and checksums.
It is also claimed that FL does not solve the "basic issue that training ... requires data transmission to cloud resources". What does this mean? You obviously need to communicate something to learn anything, and the point in FL is to send not raw data but e.g. gradients or model weights. This still does not protect privacy (for that you would use something like differential privacy), but it does generally make attacking the system harder.
I also have no idea how the adverse business ramification works, but this is a minor issue.

6) General comment on citations: many of the citations are very strange. For example, the standard reference for MNIST data is LeCun et al. 1998, who introduced the dataset, not Deng 2012. Similarly, federated learning was introduced by McMahan et al. 2016, and a nice survey is Kairouz et al. 2019. Please check all the citations to give credit where it is due.


### References:

Carlini 2020: InstaHide Disappointingly Wins Bell Labs Prize, 2nd Place (blog post), currently available from https://nicholas.carlini.com/writing/2020/instahide_disappointingly_wins_bell_labs_prize.html

Hashemi et al. 2020: DarKnight: A Data Privacy Scheme for Training and Inference of Deep Neural Networks.

Huang et al. 2020: InstaHide: Instance-hiding Schemes for Private Distributed Learning.

Kairouz et al. 2019: Advances and open problems in federated learning.

LeCun et al. 1998: Gradient-Based Learning Applied to Document Recognition.

McMahan et al. 2016: Communication-efficient learning of deep networks from decentralized data.

---

> ### Author Response · Authors · 2022-12-22
> **Response to Reviewer F9qK**
>
> We appreciate all of the reviewers’ obvious expertise in this subject area and their thoughtful responses to our manuscript.
>
> Here we respond point-by-point to the reviewer’s comments, with the reviewer comments in plain text and our responses [ flanked by brackets ].
>
> Weaknesses: There are ...
> [  It was clear that we needed to greatly improve our description of the privacy definition and threat model, as requested by this reviewer and the other reviewers. We have added a new section ‘Threat model’ with an accompanying new Figure 1 to more clearly delineate and discuss the threat model and privacy implications, and our Results and Discussion sections now better address the information leakage and privacy implications of our encoding process in the specific context of this threat model, with addition of new subsections 5.7 ‘Privacy analysis’, 5.8 ‘Information leakage’, and new Figures 9, 10, & 11. As described in detail our revised manuscript, the privacy goal is not to protect the label of an image (indeed, it is assumed in the threat model that an attacker would have access to all labels of encoded training images and could obtain access to labels of uncertain quality for unencoded images that they might present at inference). Instead, the goal is to obfuscate the original images, given an attacker that possesses the encoded images and associated labels. ]
>
> Related to ...
> [  Our additions to the manuscript of the new section 2 ‘Threat model’ with the accompanying new Figure 1, together with the reworked privacy analysis and discussion (5.7 ‘Privacy analysis’,  5.8 ‘Information leakage’, and new Figures 9, 10, & 11), serve to better define the threat model and privacy claims. We have improved our Results and Discussion sections to better address these limitations of the original version, adding a clear definition of the threat model, the associated information leakage, and a detailed analysis showing, both theoretically and via analysis, that the information leaked (the association between encoded examples and labels) does not simplify the problem of attempting to reverse the label-agnostic random functions that are applied between original and encoded images. ]
>
> For most ...
> [  We have added much more clarifying language that in the current paper, we only test the accuracy of our encoding methods in the context of fully-connected ANNs, and that integration of the method with convolutional neural networks is beyond the scope of this initial report. ]
>
> Related to 3)...
> [  Based on the feedback from this reviewer, as well as similar comments from the other reviewers, we have now added training data from CIFAR-10 and Fashion-MNIST, as well as results from training on our synthetic unknown challenge dataset. These data have been added to a new section 5.6 ‘Testing on alternative data sources’ and a new Table 1 showing the absolute decrease in validation accuracy from original to encoded data across these datasets. ]
>
> The alternative ...
> [  As discussed in the responses to the other reviewers, we have added references and new discussion subsections corresponding to the InstaHide and NeuraCrypt approaches, together with the Carlini, et al., attacks on each of these methods, and also the Hashemi, et al. DarKnight approach.  ]
>
> 5.2) Some of the claims...
> [  We hope that our revised threat model and privacy discussion and analysis clarify some of these issues, and we also have clarified this point regarding federated learning in the revised manuscript. While FL approaches should only transmit data such as updated weights, the concern that a data owner might have with this approach is that the other party’s software does have direct access to the unencoded data. Our previous statement about requiring data transmission to cloud resources was trying, perhaps inelegantly, to say that initial training on large datasets can be beyond the often lightweight local resources used in many distributed FL projects. For the sake of clarity, we have deleted this sentence.  ]
>
> General comment...
> [ We have revised our citations accordingly and included the above references. ]
>
> Requested Changes: ...
> [ As described in detail above, we have substantially improved the paper based on reviewer feedback to address these issues, including addition of Section 2 ‘Threat model’, 5.6 ‘Testing on alternative data sources’, 5.7 ‘Privacy analysis’,  5.8 ‘Information leakage’, and new Figures 1, 9, 10, & 11, a reworked discussion subsection 6.2 ‘Security aspects of the method’ with new subsubsection 6.2.1 ‘Security parameters’, to emphasize the security parameters in the method, as well as a new Table 1. ]

---

> > ### Comment · Reviewer_F9qK · 2023-01-11
> > **Final comment**
> >
> > Thanks you for making the effort, unfortunately, I am still not convinced, as detailed below.
> >
> > ## Update for the fixed paper
> >
> > While I think the paper is now better in many ways, I still do not think it is ready for publishing. My main concerns are the following:
> > 1) I still think the security notion in the paper is simply too hazy to warrant the claims the authors make: in Section 5.7, what does it mean that the adversary cannot find a close approximation to the encoding function? Does this protect the encoding function or the input? Is it a breach if I can reconstruct, say, a recognizable face in an input image, but the colors are off and I cannot tell where in the original image the face is? This issue is made worse by various statements letting the reader understand that the method provides some strict security and privacy notions (compare e.g. to the abstract).
> >
> > 2) The security proof is too hand wavy (see Section 5.7).
> >
> > 3) On the security parameters in Section 6.2.1, this is now simply a list of all hyperparameters and design choices. These would surely jointly set the hardness of the problem, but what I would interpret as a security parameter is something that unambiguously sets the hardness of the problem when you change the value, e.g., key length for standard encryption. If you here consider e.g. location of the padding included in the list: how would this change the hardness of the problem? In case I shuffle the pixel values afterwards, what effect would this have?
> >
> > ### Less important issues:
> >
> > i) It is very hard for me to see that all the assumptions about the adversary could plausibly hold in any real deployment setting. However, I do not consider this as a major issue preventing acceptance, because these are at least stated plainly, and any potential user of the method hopefully understands how strict assumption it is that, e.g., the input size of images is considered to be sensitive information that should not leak beyond the trusted parties, or that basically any user submitting images to the decentralised system has accesss to the same encoding function and can therefore decrypt anyone else's encoded images.
> >
> > ii) User privacy and secure computation are not the same thing, please do not use them as synonyms.

---

### Decision · Action_Editors · 2023-01-15

**Recommendation:** Reject

**Comment:**

This paper studied neural network training on security-enhanced datasets. While the reviewers agree on some merits and technical insights proposed in this paper, there are several major concerns such as the threat model, the security proof, and the hyperparameter selection raised by reviewers. The authors' response and updated version alleviate some points but do not address the concerns fully. The reviewers have provided their post-rebuttal comments and suggestions to improve this work.

Based on the reviewers' comments and my own evaluation, I decided to recommend rejecting this submission in its current form. I hope the authors find the review comments useful to improve future versions.

**Audience:**

May not be of top interest to the general TMLR audience

**Claims And Evidence:**

The claims and evidence are somewhat limited and require further justification and comparisons